# Distinct neutralization sensitivity between adult and infant transmitted/founder HIV-1 subtype C viruses to broadly neutralizing monoclonal antibodies

Bongiwe Ndlovu[1ʘ], Kamini Gounder[1,2,3ʘ], Nelisiwe Zikhali[1ʘ], Lanish Singh[1,2], Ntokozo Ntshangase[1], Nombali Gumede[1], Jane Millar[1,4], Rebecca T. van Dorsten[5], Nicholas E. Grayson[4], David Bonsall[6], Sandra E. Chaudron[6], Jennifer Mabuka[2], Krista L. Dong[7], Bruce D. Walker[1,7,8], Penny L. Moore[5,9,10], Philip J.R. Goulder[1,2,3,4], Thumbi Ndung'u[1,2,3,7]*

1 HIV Pathogenesis Programme, The Doris Duke Medical Research Institute, University of KwaZulu-Natal, Durban, South Africa, 2 Africa Health Research Institute, Durban, South Africa, 3 Division of Infection and Immunity, University College London, London, United Kingdom, 4 Department of Paediatrics, Nuffield Department of Medicine, University of Oxford, Oxford, United Kingdom, 5 Centre for HIV and STIs, National Institute for Communicable Diseases of the National Health Laboratory Service, Johannesburg, South Africa, 6 Wellcome Centre for Human Genetics, University of Oxford, Oxford, United Kingdom, 7 Ragon Institute of Massachusetts General Hospital, Massachusetts Institute of Technology and Harvard University, Cambridge, Massachusetts, United States of America, 8 Howard Hughes Medical Institute, Chevy Chase, Maryland, United States of America, 9 South African Medical Research Council (SAMRC) Antibody Immunity Research Unit, School of Pathology, Faculty of Health Sciences, University of the Witwatersrand, Parktown, Johannesburg, South Africa, 10 Centre for the AIDS Programme of Research in South Africa (CAPRISA), University of KwaZulu-Natal, Durban, South Africa

ʘ These authors contributed equally to this work.
* thumbi.ndungu@ahri.org

## Abstract

Broadly neutralizing antibodies (bnAbs), passively administered or elicited through vaccination, are a promising strategy for novel HIV prevention, treatment or inducing ART-free remission. However, HIV diversity and evolution are a barrier to the efficacy of bnAbs and there is therefore an urgent need for continuous virus surveillance to identify bnAbs with optimal neutralization breadth and potency against transmitted/founder (TF) viruses, especially in high-burden regions. We determined the neutralization sensitivity of TF viruses isolated within seven days after first detection of heterosexually acquired infection from young women 18–23 years old (n = 39) and within 1 month after birth from in-utero infected infants (n = 21) from FRESH and Baby Cure cohorts respectively, in KwaZulu-Natal, South Africa, where HIV-1 subtype C predominates. Neutralization sensitivities of 47 viruses from FRESH and 21 viruses from Baby Cure were assessed against nine bnAbs targeting different regions on the HIV-1 Env trimer. HIV-1 *env* sequences within and between bnAb epitopes were compared with database. The bnAbs VRC07–523LS, CAP256-VRC26.25, PGDM1400, 10E8 and PGT151 displayed higher neutralization breadth and potency than other bnAbs against FRESH TF viruses

**Data availability statement:** Neutralization data is available in the supporting information files. HIV-1 env nucleotide sequences are available in GenBank database. Accession numbers for the FRESH cohort sequences range from PQ874248-PQ874674, while accession numbers for the BABY Cure cohort are range from PV016540-PV016586. These sequences are now available to the public.

**Funding:** This study was supported in part by grants from the Bill and Melinda Gates Foundation (OPP1066973 and OPP1146433 to TN, INV-055901 to BN). European & Developing Countries Clinical Trials Partnership (TMA2020CDF-3183 to BN), National Research Foundation Thuthuka Program (129653 to BN), Gilead Sciences, Inc. (00406 to TN), the International AIDS Vaccine Initiative (IAVI) (UKZNRSA1001 to TN), the NIAID (R37AI067073 to BDW), the Witten Family Foundation, the Dan and Marjorie Sullivan Foundation, the Mark and Lisa Schwartz Foundation, Ursula Brunner, the AIDS Healthcare Foundation, and the Harvard University Center for AIDS Research (HU CFAR NIH/NIAID, P30 AI060354 to BDW and 5P30AI060354-15 to KG), which is supported by the following institutes and centers co-funded by and participating with the US National Institutes of Health: NIAID, NCI, NICHD, NHLBI, NIDA, NIMH, NIA, FIC, and OAR. This work was also partially supported through the Sub-Saharan African Network for TB/HIV Research Excellence (SANTHE) which is funded by the Science for Africa Foundation to the Developing Excellence in Leadership, Training and Science in Africa (DELTAS Africa) programme [Del22-007 to TN] with support from Wellcome Trust and the UK Foreign, Commonwealth & Development Office and is part of the EDCPT2 programme supported by the European Union; the Bill & Melinda Gates Foundation [INV-033558 to TN]; and Gilead Sciences Inc., [19275 to TN]. All content contained within is that of the authors, the funders had no role in the study design, data collection and analysis, decision to publish or preparation of the manuscript.

**Competing interests:** The authors have declared that no competing interests exist.

(>70% coverage, starting concentration of 10 µg/ml). Furthermore, VRC07–523LS showed higher neutralization breadth and potency than other bnAbs against Baby Cure TF viruses (p = 0.02). Interestingly, CAP256-VRC26.25 and PGT151 had lower neutralization coverage against infant TF viruses (<60% coverage). Moreover, 40% of infants TF had escape mutations within the V2 loop compared to 28% observed in FRESH and these mutations may explain the observed differences in neutralization sensitivities. However, few mutations were observed in gp120-gp41 interface in both adults and infants. Our findings suggest that intervention studies may have to consider different antibody combinations in adult versus paediatric settings. Moreover, high transmission of escape variants in both vertical and heterosexual transmissions is of concern. This information may be important in the selection of bnAbs that will undergo clinical testing in subtype C settings.

## Author summary

There is an urgent need for alternative prevention and treatment strategies including vaccines and passively administered broadly neutralizing antibodies (bnAbs). Passive administration of bnAbs has the ability to prevent infection and delay viral rebound in humans and non-human primates. However, evidence suggests that HIV-1 is becoming more resistant to bnAbs over time, highlighting the need for ongoing viral surveillance. Characterization of recently transmitted/founder viruses may provide useful information that informs selection of bnAbs for evaluation in human clinical trials. This study determined differences in neutralization sensitivity and amino acid variations between adult and infant recently transmitted viruses (transmitted/founder viruses). Adult-derived transmitted/founder (TF) viruses had high sensitivity to the combination of VRC07–523LS, CAP256-VRC26.25 and 10–1074 or PGT151. In contrast, infant-derived TF viruses had high sensitivity to the combination of VRC07–523LS, 10–1074 and 10E8. Almost half of the infant viruses had resistance mutations in the V2 apex contact residues, while adults had a high frequency of escape mutations in the V3 loop. This study suggests that different bnAb combinations should be considered in prevention or treatment of adult and paediatric populations.

## Introduction

HIV-1 infection remains a global public health challenge. In 2023, approximately 39.0 million [33.1 - 45.7 million] people were living with HIV-1 and 630 000 [480 000–880 000] AIDS-related deaths were reported despite the availability of effective treatment and prevention tools [1]. In sub-Saharan Africa, new infections are disproportionately high among adolescent girls and young women between the age of 15–24 years [1]. Current prevention strategies including the use of oral pre-exposure prophylaxis (PrEP) or long-term injectables are highly effective in reducing the risk of HIV-1

acquisition [2–5]. However, there are significant challenges associated with these tools including stigma, poor adherence, variable access, drug toxicity and side effects, particularly in children where drugs are not easy to take. This highlights the need for alternative preventive or therapeutic strategies such as vaccines or passively administered broadly neutralizing antibodies (bnAbs), with the latter showing promising results in both preclinical and clinical studies [6–12].

Passive administration of bnAbs prevented simian-human immunodeficiency virus (SHIV) infection and delayed viral rebound in non-human primates [6–9,13]. In humanized mice, bnAbs temporarily suppressed HIV-1 viremia, with viral load rebound observed after a few weeks of monotherapy due to the emergence of escape variants [14–17]. Several bnAbs have subsequently undergone evaluation in human clinical trials and were shown to be safe and effective in delaying viral rebound following antiretroviral treatment (ART) interruption in chronically treated HIV-1 infection [10–12,18–22]. Recently, the antibody mediated prevention (AMP) trials (HVTN 704/HPTN 085 and HVTN 703/HPTN081) provided a proof of concept that bnAbs can prevent HIV-1 infection, with VRC01 providing partial protection limited to VRC01-sensitive strains [23]. The AMP trial validated bnAbs as a mechanism of protection against HIV acquisition and further supports to the efforts to design immunogens capable of eliciting such bnAbs. However, further exploration is required to ensure that efforts focusing on bnAbs are likely to be most successful against recently transmitted viruses in diverse settings and to enhance the design of antibodies for HIV prevention and immunotherapy.

BnAbs targeting the CD4 binding site, V2-apex, V3-glycan supersite and the membrane proximal external region (MPER) of gp41 were tested against 200 acute or early chronic subtype C viruses; VRC07–523LS, CAP256-VRC26.25, 10-1074V (a variant of the parental 10–1074) and 10E8 had the highest neutralization breadth and potency [24]. Similarly, VRC07–523LS and 10E8v4 had the greatest neutralization breadth against subtype C viruses, whereas high neutralization potency was observed with CAP256-VRC26.25 and PGDM1400 [25]. Additional studies indicated that a combination of bnAbs that target different epitopes might have improved neutralization breadth and potency [26,27]. Computational prediction tools suggested that the most effective combination against subtype C viruses involved CAP256-VRC26.25, 10-1074V and VRC07–523LS [26]. Taken together, these studies identified the best individual and combination bnAbs that may provide the optimal antiviral neutralization coverage against subtype C viruses, albeit in adult participants only.

However, there is evidence that HIV-1 is becoming more resistant to bnAbs over the years of the epidemic [25,28,29]. The growing body of evidence indicate that currently circulating subtype C viruses have increased resistance to some bnAbs compared to historical viruses and this was shown in both African and Indian viruses [25,28,29] and similar findings have been reported in subtype B [30]. Mechanisms that mediate resistance to bnAbs are complex and may include the deletion of N-linked glycans in contact positions, single-point mutations within bnAb epitopes and elongation of the V1 loop [31–35]. Thus, ongoing virus surveillance efforts are important to ensure that current prevention strategies remain relevant to circulating viruses. Furthermore, despite progress, vertical transmission remains unacceptably high particularly in southern Africa, and significant differences may exist between heterosexual and vertical transmission due to different routes of exposure [36]. *In utero*-transmission of HIV occurs via the placenta, evidence suggests that cell-associated virus is transmitted when trophoblasts (epithelial layer in the placenta) fuse with maternal lymphocytes [37–39]. Moreover, infants are exposed to maternal antibodies in utero, which could select for TF Env variants that are resistant to some bnAbs. Cell-to-cell transmission may also contribute to breakthrough infections because of high bnAb resistance in cell-to-cell compared to cell free neutralization [40–42]. The dynamics of bnAb development also differ between children and adults [43–45], and although the mechanisms are not fully understood, these differences could be partially related to different characteristics of TF viruses between infants and adults.

Here we cloned the full-length *env* from plasma samples collected from the Females Rising through Education, Support and Health (FRESH) and Ucwaningo Lwabantwana (Baby Cure) cohorts [46–48] at the earliest available time points following transmission. We characterized the adult FRESH and Baby Cure TF subtype C viruses and assessed their sensitivity to bnAbs that target the CD4 binding site, V2-apex, V3-glycan supersite, membrane proximal external region of gp41 and gp120-gp41 interface. These bnAbs are currently under evaluation in clinical trials, and these data would allow us to

identify bnAbs likely to be effective in adult versus infant HIV prevention or immunotherapeutic strategies such as passive immunization, vaccine design and HIV cure interventions. We also analysed the genetic sequence differences that may discriminate between adult and infant T/F Env and between viruses that are sensitive or resistant to various bnAbs.

## Materials and methods

### Ethics statement

All subjects provided written informed consent to participate in protocols that were approved by the Biomedical Research Ethics Committee (BREC) of the University of KwaZulu-Natal and Institutional Review Board of Massachusetts General Hospital (BF131/11 and 2012-P001812). Legal guardians of the infants provided written informed consent to participate in the study and the Biomedical Research Ethics Committee of the University of KwaZulu-Natal and Oxfordshire Research Ethics Committee approved these studies.

### Study population

Plasma samples were collected from thirty-nine women living with acute HIV infection from the Females Rising through Education, Support and Health (FRESH) cohort from 2013 to 2020, and twenty-one infants living with *in-utero* transmitted HIV from the Ucwaningo Lwabantwana (Baby Cure) study from 2015-2020 [46–48]. Briefly, the FRESH cohort is an observational, prospective acute HIV infection cohort in Umlazi, South Africa [46]. Young women between 18–23 years old who were sexually active and at high risk for HIV-1 infection were enrolled in a socioeconomic empowerment and life skills building programme that incorporates twice weekly screening for emergent HIV-1 infection using HIV-1 RNA PCR (Nuclisens EasyQ v2.0 assay, Biomeriuex, Marcy I'Etoile, Switzerland) [46]. Plasma viral load and CD4$^+$ T-cell counts were measured longitudinally in all study visits.

The Baby Cure study is an ongoing observational study investigating the role of early antiretroviral treatment (ART) initiation in South African HIV-1 infected infants as previously described [48]. Mothers were recruited following delivery after the diagnosis of HIV infection had been made in the child. All infants born to HIV seropositive mothers received nevirapine from birth and an additional 70% of those infants enrolled additionally received zidovudine (AZT) twice a day as prophylaxis against intra-partum vertical transmission. The infants were tested for HIV infection via DNA PCR using dried blood spots. Additional tests were point of care testing using total nucleic acid PCR (GeneXpert Qualitative HIV-1 PCR, Cepheid, Sunnyvale, CA, USA) and HIV RNA plasma RNA viral load quantification (Nuclisens EasyQv2.0, BioMerieux, I'Etoile, France). Infants who tested positive for HIV-1 on two or more separate nucleic acid tests were initiated on ART including zidovudine (AZT), lamivudine (3TC) and nevirapine (NVP) upon diagnosis. Blood samples were collected within 48 hours after birth and monthly following a positive PCR result. Plasma viral load was measured longitudinally using the Nuclisens HIV-1 RNA PCR assay. Meanwhile, CD4$^+$ T cells were measured using Tru-count technology and analysed further with flow cytometry according to the manufacturer's instructions (Becton Dickson, BD Biosciences, San Jose, CA, USA).

### Viral RNA extraction and HIV-1 Env single genome amplification

In the FRESH cohort, HIV-1 viral RNA was extracted from plasma samples collected a median of 1 day after the first detection of viremia. In the Baby Cure cohort, viral RNA was extracted from plasma samples collected at birth or one month of infection. HIV-1 RNA was extracted using the QiaAmp Viral RNA Mini kit (Qiagen, Hilden, Germany) and eluted in elution buffer as per manufacturer instructions. Viral RNA was converted into cDNA using Superscript IV reverse transcriptase enzyme (Invitrogen, Carlsbad, CA, USA) according to the manufacturer's instructions with the specific primer, OFM19 (5′-GCACTCAAGGCAAGCTTTATTGAGGCTTA-3′).

Thereafter, full-length HIV-1 *env* gene was amplified using a limiting endpoint dilution as previously described [43]. Briefly, the cDNA was serially diluted in eight replicates of PCR reactions and amplified using nested PCR. The dilution

that yielded approximately 30% of positive reactions were selected for a second round of PCR to increase the likelihood of amplifying from a single template. The first round of PCR was carried out using the High Fidelity Platinum enzyme (Invitrogen, Carlsbad, CA, USA) and the external *env* primers VIF1 (5′-GGGTTTATTACAGGGACAGCAGAG-3′) and OFM19 (5'-GCACTCAAGGCAAGCTTTATTGAGGCTTA-3'). The second round of PCR was performed using the Phusion High Fidelity polymerase (Thermo Fisher, Waltham, MA, USA) together with the ENV A (5′-GCTTAGGCATCTCCTATGGCAG GAAGAA-3′) and ENV N (5′-CTGCCAATCAGGGAAGTAGCCTTGTGT-3′) primers. PCR products were analysed on 1% agarose gel and gel purified using the Illustra GFX PCR DNA and Gel Band Purification Kit (GE Healthcare, Pittsburgh, PA, USA).

## Sanger sequencing

All single genomes generated were sequenced directly using the ABI Big Dye Terminator V3.1 cycle sequencing kit (Applied Biosystems, Foster City, CA, USA). Overlapping DNA fragments were assembled and edited using the Sequencher software program version 5.4.6 (Gene Codes Corporation, Michigan, USA) and were aligned using Clustal W. Maximum-likelihood phylogenetic trees were constructed using Geneious version 2022.0.1 (Biomatters, New Zealand) to assess the relatedness of the Env sequences and reference strains. Subtype reference strains were obtained from the Los Alamos HIV sequence database (https://www.hiv.lanl.gov/content/sequence/NEWALIGN/align.html).

## Intrapatient diversity

Intrapatient diversity scores were determined by calculating the average pairwise nucleotide differences among sequences obtained from each participant as previously described [49]. Sequences were aligned and the average number of nucleotide base substitutions per site between all sequence pairs was determined using MEGA software. The average pairwise distance represented the intrapatient diversity scores and intrapatient diversity percentage was calculated. In addition, sequences were aligned using ClustalW and Highlighter plots were generated using the Los Alamos National Laboratory HIV Sequence Database Highlighter tool (https://www.hiv.lanl.gov/content/sequence/HIGHLIGHT/highlighter_top.html).

## Next-generation sequencing

HIV-1 *env* from mother and infant plasma was reverse transcribed and amplified. Total RNA was extracted using the NucliSens easyMAG system (bioMérieux) and concentrated with Agencourt RNAClean XP (Beckman Coulter, Pasadena, CA, USA). Libraries retaining directionality were prepared using the SMARTer Stranded Total RNA-Seq Kit v2 - Pico Input Mammalian (Takara Bio, Kyoto, Japan) with specific modifications to the manufacturer protocol as previously described [50]. Deep sequencing was then performed using an Illumina MiSeq machine (Illumina, San Diego, CA, USA) and a PacBio sequencing machine (Pacific Biosciences, Menlo Park, CA, USA). Raw reads were blasted against the Los Alamos Sequence Database (http://www.hiv.lanl.gov/) HIV-1 "Web Alignments 2017" to obtain the HXB2 alignment positions for each read. From the aligned sequences, overlapping k-mers were generated to construct the genome via a De Bruijn graph.

## Transmitted/founder viruses

Transmitted/founder viruses are virus strains that are transmitted and establish an infection in a new host. Previous studies have demonstrated that both heterosexual and vertical HIV transmissions are characterized by a transmission genetic bottleneck, such that in the majority of cases, a new infection is established by a single or very limited number of closely related viral genetic variants, despite the presence of diverse quasispecies in the donor [51–54]. Participants infected with a single strain or closely related strains have homogenous viral sequences during acute infection, while participants

infected with multiple diverse variants have heterogeneous viral sequences [51,55]. In this study, samples were collected at birth in the Baby Cure and in the first week of infection in the FRESH cohort (with a negative HIV-1 RNA test within the last 4 days) indicating that these represent TF or very closely related viruses. HIV *env* SGA sequences were analysed to determine the TF sequence as previously described [56]. Briefly, HIV-1 *env* SGA sequences were analysed; majority of the participants had homogenous sequences indicative of a single variant infection, while a few participants showed only a limited number of TF variants. For participants with HIV-1 *env* sequences indicative of a single TF variant, a single representative sequence was selected to represent the viral population circulating in the participant. In cases with viral sequence populations indicative of more than one TF variant, two *env* sequences were selected to capture the viral diversity in the FRESH participants. However, in the Baby Cure cohort, due to sample volume limitation that precluded a limiting dilution approach for analysis of Env diversity, a single bulk amplified clonal sequence was selected from each infant.

## Generation of *env* clones

Purified amplicons were cloned into the eukaryotic pcDNA 3.1 Directional TOPO expression vector, which was then used to transform TOP 10 *E. coli* competent cells (Thermo Fisher, Waltham, MA, USA) as previously described [57,58]. *Env* plasmid DNA was isolated from bacterial cultures using GeneJET Plasmid Miniprep kit (ThermoFisher Scientific Inc., USA). Some of the *env* clones were produced using codon optimization, whereby full-length *env* sequences were codon optimized and then cloned into pcDNA 3.1 (GenScript Biotech, Rijswijk, Netherlands). *Env* clones were sequenced by Sanger sequencing and the sequences were aligned with the original SGA sequence to ensure that there were no amino acid differences between the original and codon optimized sequences.

## Production of HIV-1 Env pseudoviruses

Env-pseudotyped viruses were prepared by co-transfection of HIV-1 TF *env* with HIV-1 *env*-defective backbone plasmid (pSG3Δ*env*) and X-tremeGENE HP DNA Transfection Reagent (Roche Diagnostics, Mannheim, Germany) in HEK293 T cells (ATCC, Manassas, VA, USA) as previously described [35,58]. After the 48-hour incubation, the supernatant containing pseudoviruses was harvested, filtered through a 0.45 µm filter and stored at -80°C.

## Determination of TCID$_{50}$

Viral infective dose (TCID$_{50}$) was assessed in TZM-bl cells (NIH AIDS Research, Reference Reagent Program). Viral stock was serially diluted in four-fold dilutions (starting dilution 1:4) in 96-well flat bottom plates (Sigma Aldrich, Darmstadt, Germany). Freshly trypsinized TZM-bl cells (10 000 cells/well) containing dextran-hydrochloride (DEAE) (Sigma-Aldrich, St Louis, MO, USA) were added to the wells and the plates were incubated at 37°C for 48 hours. After incubation, viral infectivity was measured by adding Bright-Glo luciferase reagent (Promega Madison, Wisconsin, United States) and the plates were read using a PerkinElmer luminometer (PerkinElmer Life Sciences, Separation Scientific, Model Victor Nivo). A negative control of TZM-bl cells without the virus was included as a negative control in each experiment.

## Neutralization assay

The neutralization activity was measured as a reduction of virus infection in TZM-bl cells after a single round of infection using Tat-regulated firefly luciferase (Luc) reporter gene expression as previously described [59]. The sensitivity of the plasma-derived transmitted/founder Env variants generated from infants and adults to nine bnAbs that target the CD4 binding site (VRC01, VRC07–523LS and 3BNC117), V2 apex (CAP256-VRC26.25, PGDM1400), V3-glycan supersite (PGT121, 10–1074), MPER (10E8) and gp120-gp41 interface (PGT151) were measured. Briefly, the nine bnAbs (starting concentration 10 µg/ml) were serially diluted in 3-fold sequence in a 96-well plate. Diluted bnAbs were incubated with single-round competent TF pseudoviruses for 1 hour at 37°C. After incubation, freshly trypsinized TZM-bl cells containing

dextran-hydrochloride (DEAE) were added into the 96-well plate and incubated at 37°C for 48h. After incubation, the cells were lysed using Bright-Glo Luciferase system (Promega Madison, Wisconsin, United States) and the neutralization activity was measured on a PerkinElmer microtiter plate luminometer and reported as Relative Light Units (RLUs). TZM-bl cells without the virus were used as a negative control, while the transmitted/founder virus incubated with the TZM-bl cells were used as a positive control. The concentration of the antibody where the infection (relative light units) was reduced by 50% ($IC_{50}$ titers) compared to the virus control wells was calculated for each monoclonal antibody.

## Statistical analysis

The difference in viral load or neutralization sensitivity between infant and adult TF viruses were determined by comparing the median viral load and $IC_{50}$ titers using Mann Whitney t-test. A p-value <0.05 indicates significant differences between the groups.

## Results

### Clinical characteristics and viral loads of the study participants

Plasma samples were obtained from thirty-nine women living with HIV-1 subtype C from the FRESH cohort at a median of one day post first observed plasma viremia (Table 1). The study participants were between 18–23 years old, Fiebig stage I-V at the time of diagnosis (Table 1). In addition, twenty-one infants born with HIV from the Baby Cure study were selected from a cohort of 151 in-utero infected infants based on the availability of blood samples [48]. Fifteen infants were females, while six infants were males (Table 2). Blood samples were collected within 48 hours after birth for sixteen infants and one month after birth for four infants.

The median absolute CD4 + T-cell count was 631 cells/µL in the FRESH cohort (Table 1). In Baby Cure cohort, the median absolute CD4 + T-cell count was 1,952 cells/µL and the median CD4% was 38% (Table 2). The median viral loads in both the FRESH and Baby Cure cohorts at the time of analysis were 59,000 RNA copies/ml (Fig 1A). At birth, the mothers for infants in the Baby Cure cohort had a median VL of 8,600 copies/ml, and a median CD4$^+$ T-cell count of 558 cells/µL. The median viral load was significantly higher in FRESH participants compared to mothers in the Baby Cure cohort (p < 0,001) (Fig 1B), consistent with the fact that >90% of mothers were receiving ART at the time of delivery compared to 0% of the FRESH cohort participants at this time point. Furthermore, infants VL was higher than their transmitting mothers (p = 0,02) (Fig 1C). There was no significant difference in CD4$^+$ T cell counts between FRESH and Baby Cure mothers (Fig 1D).

### Low level of sequence diversity of HIV-1 subtype C transmitted/founder *env* in adults and infants

To determine the level of genetic diversity of the TF virus in the FRESH cohort, 383 single HIV-1 env genome sequences were generated from plasma collected between 1–8 days post-infection from 39 participants (Fig 2A). The median number of full-length HIV-1 *env* SGA sequences generated was six per participant. Maximum-likelihood phylogenetic trees, intrapatient diversity and Highlighter Plots were constructed using the HIV-1 *env* sequences generated (S1 Fig and S1 Table). Most of the FRESH study participants were infected with subtype C viruses except for participant 527 who was infected with an AC recombinant (Fig 2A). As expected, the majority of viral sequence populations generated from the acute phase of infection before peak viremia were highly homogeneous suggesting that most of the study participants were infected with a single variant. Overall, these participants (PIDs: 036, 039, 093, 186 and 318) also exhibited very low intrapatient viral diversity, consistent with infection by a single or closely related founder virus population (S1 Fig and S1 Table). However, seven participants (201, 271, 451, 498, 922, 1952 and 2148) displayed at least two distinct clusters of viral populations suggestive of infection with multiple variants (Fig 2A). These participants had slightly high intrapatient diversity suggesting infection with more than one viral variant (S1 Table and S1 Fig). However, SGAs representative of

**Table 1. Clinical characteristics of the FRESH study participants including the number of days post first observed plasma viraemia for the sample analysed for this study, the stages of early infection when the blood samples were collected, viral load and CD4 T-cell count at the time of sample collection.**

| Participant ID | DPOPV[a] | Fiebig stage[b] | Viral Load (copies/ml) | CD4 count (cells/mm³) |
|---|---|---|---|---|
| 036 | 3 | II | 2 400 000 | 204 |
| 039 | 1 | III | 400 000 | 432 |
| 079 | 3 | II | 240 000 | 637 |
| 093 | 3 | I | 31 000 | 493 |
| 102 | 3 | III | 270 000 | 814 |
| 186 | 4 | III | 6 300 000 | 306 |
| 198 | 1 | III | 3 700 000 | 367 |
| 201 | 1 | IV | 760 000 | 474 |
| 208 | 3 | II | 46 000 | 390 |
| 267 | 7 | III | 210 000 | 399 |
| 268 | 4 | II | 95 000 | 457 |
| 271 | 8 | III | 9 850 000 | 458 |
| 272 | 1 | II | 87 000 | 774 |
| 318 | 1 | II | 160 000 | 911 |
| 451 | 1 | V | 62 000 | 434 |
| 479 | 1 | I | 14 000 | 1025 |
| 498 | 1 | I | 54 000 | 631 |
| 499 | 1 | I | 53 000 | 738 |
| 519 | 1 | I | 120 000 | 802 |
| 527 | 2 | III | 38 000 | 685 |
| 559 | 1 | I | 2 200 | 980 |
| 594 | 3 | I | 30 000 | 453 |
| 627 | 1 | I | 17 000 | 921 |
| 651 | 2 | I | 3 900 | 851 |
| 726 | 2 | I | 23 000 | 873 |
| 857 | 1 | I | 1 400 | 874 |
| 920 | 1 | I | 15 000 | 685 |
| 922 | 1 | I | 61 000 | 421 |
| 970 | 4 | I | 500 000 | 475 |
| 1074 | 1 | I | 35 000 | 525 |
| 1088 | 1 | I | 14 000 | 673 |
| 1199 | 1 | III | 1 800 000 | 355 |
| 1368 | 1 | I | 59 000 | 763 |
| 1388 | 1 | II | 14 000 | 714 |
| 1439 | 1 | I | 72 000 | 863 |
| 1512 | 1 | I | 17 000 | 623 |
| 1685 | 1 | II | 99 000 | 341 |
| 1952 | 1 | I | 37 000 | 1211 |
| 2148 | 1 | I | 12 000 | 473 |
| **Median** | **1** | **I** | **59 000** | **631** |
| | | | | |

[a]DPOPV denotes days post observed plasma viremia

[b]Stage of early infection when T/F samples were collected

Viral load of T/F sample

CD4 count of T/F sample

**Table 2. Clinical characteristics of the Baby Cure study participants including the time points of sample collection, viral load, CD4⁺ T-cell count and CD4⁺ T-cell % and transmitting mother's viral load and CD4⁺ T cell count.**

| INFANTS | | | | | MOTHERS | | |
|---|---|---|---|---|---|---|---|
| Participant ID | Time point | Viral Load (copies/ml) | CD4% | CD4 count (cells/mm³) | Participant ID | Viral Load (copies/ml) | CD4 count (cells/mm³) |
| 0048_1 | Birth | 4,300,000 | 38 | 831 | 0048_0 | 210,000 | 5 |
| 0180_1 | Birth | 35,000 | 46 | 2444 | 0180_0 | 22,000 | 365 |
| 0204_1 | Birth | 650,000 | 51 | 4084 | 0204_0 | 250 | 446 |
| 0215_1 | 1 month | 2,600,000 | 27 | 2224 | 0215_0 | 180 | 877 |
| 0239_1 | Birth | 8,500,000 | 23 | 500 | 0239_0 | 27,000 | 509 |
| 0758_1 | Birth | 4,500 | 33 | 924 | 0758_0 | 8,600 | 241 |
| 0778_1 | Birth | 170,000 | 35 | 1680 | 0778_0 | 11,000 | 707 |
| 0902_1 | Birth | 45,000 | 44 | 3129 | 0902_0 | 3,400 | 1201 |
| 1778_1 | Birth | 40,000 | 37 | 2397 | 1778_0 | 4,800 | 1081 |
| 2178_1 | 1 month | 88,000 | 22 | 642 | 2178_0 | 1,400,000 | 369 |
| 2285_1 | Birth | 6,000 | 35 | 3498 | 2285_0 | 930 | 254 |
| 4063_1 | Birth | 3,200,000 | 38 | 1952 | 4063_0 | 60,000 | 350 |
| 4082_1 | Birth | 47,000 | 28 | 717 | 4082_0 | 18,000 | 350 |
| 5055_1 | Birth | 70,000 | 55 | 2114 | 5055_0 | 190,000 | 588 |
| 6027_1 | 1 month | 59,000 | 24 | 1748 | 6027_0 | 15,000 | 558 |
| 6041_1 | Birth | 20,000 | 43 | 1304 | 6041_0 | 4,400 | 611 |
| 6082_1 | Birth | 1,800,000 | 69 | 2272 | 6082_0 | 160,000 | 407 |
| 8084_1 | Birth | 420,000 | 42 | 2212 | 8084_0 | 190,000 | 652 |
| 8171_1 | 1 month | 8,200 | 37 | 1720 | 8171_0 | 5,200 | 1736 |
| 8195_1 | Birth | 2,600 | 51 | 1625 | 8195_0 | 2,800 | 576 |
| 8277_1 | Birth | 39,000 | 51 | 4486 | 8277_0 | 210 | 693 |
| **Median** | | **59000** | **38** | **1952** | | **8600** | **558** |

All infants received the combination of AZT (Zidovudine), Lamivudine (3TC) and Nevarapine (NVP).

All mothers received the combination of Efavirenz (EFV), Tenofovir disoproxil fumerate (TDF) and Emtricitabine (FTC).

both variants were cloned into pcDNA3.1 directional TOPO expression vector and were included in both the phylogenetic tree and highlighter plot as clones A and B (Figs 2A and S1).

In contrast, due to limited sample availability, only forty-six HIV-1 T/F *env* clonal sequences were produced from bulk PCR amplicons from the twenty-one HIV-1 infected infants, with the analysed samples collected within 48 hours (n = 16) or 1 month (n = 4) infants after birth. Additional sequences were produced from seven mothers by SGA (n = 3) and next generation sequencing using Illumina MiSeq machine and PacBio sequencing platforms (n = 4) (Fig 2B). The median number of full-length HIV-1 *env* clonal sequences from each infant and each transmitting mother was one. Infant *env* sequences clustered with their transmitting mother-pair SGA or NGS sequences (Fig 2B). In addition, the majority of infants were infected with subtype C viruses except for three infants, one infant was infected with an AC recombinant (0215–1) and two infants (0204–1 and 4082–1) were infected with BC recombinants (Fig 2B). We could not determine the level of diversity or multiplicity of infection in the Baby Cure study due to inability to generate single genome amplicons with available samples.

### Differences in neutralization sensitivities between adult and infant transmitted/founder viruses

To determine the neutralization sensitivity of the TF viruses to bnAbs, a panel of nine bnAbs that target the CD4 binding site (CD4bs) (VRC01, VRC07–523LS, 3BNC117), V2-apex (CAP256-VRC26.25, PGDM1400), V3-glycan super-site (PGT121, 10–1074), MPER (10E8) and gp120-gp41 interface (PGT151) was tested against 47 HIV-1 subtype C

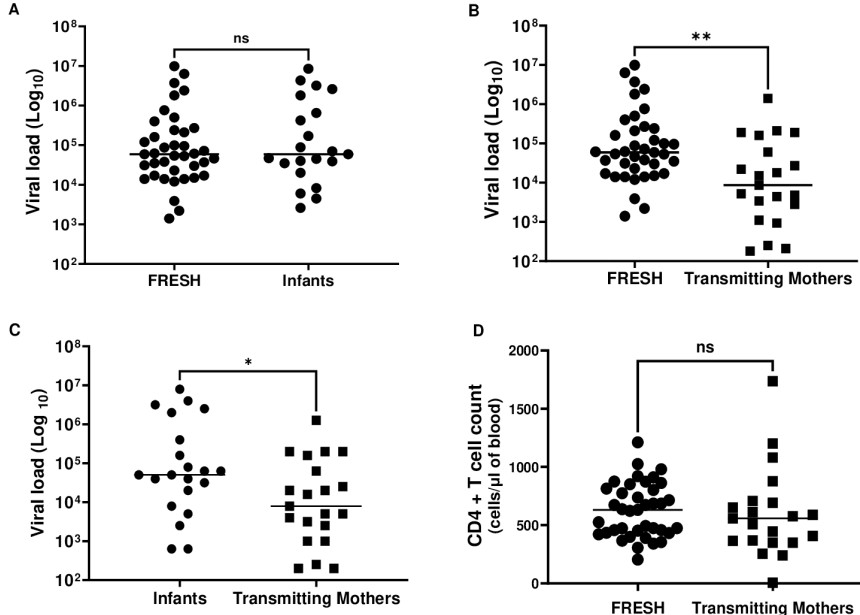

**Fig 1. The differences in viral load between women in the FRESH cohort, in-utero infected infants and their transmitting mothers from the Baby Cure cohort.** Viral load was measured in the first week of infection in the FRESH cohort, at birth or one month respectively in infants and during labour in transmitting mothers. Differences in Log viral load between FRESH participants and in-utero infected infants (A), FRESH and transmitting mothers (B), infants and transmitting mothers from Baby Cure cohort (C). Differences in CD4+ T cell counts between FRESH and transmitting mothers in Baby Cure cohort (D). Median viral load and CD4+ T cell counts were compared using Mann Whitney statistical test and * indicates p < 0,001.

TF Env-pseudotyped viruses produced from the FRESH and 21 viruses from Baby Cure studies respectively using the TZM-bl assay (Fig 3A and 3B). The neutralization activity was assessed using the percentage of viruses neutralized (coverage) and the neutralization potency (geometric mean $IC_{50}$ titers), and the data graphically illustrated in the form of heat maps (Fig 3A and 3B) and neutralization $IC_{50}$ titer scatter plots (Fig 4).

Both FRESH and Baby Cure TF viruses were highly sensitive to VRC07–523LS, this antibody neutralized 93% of TF viruses from the FRESH, whereas coverage in the Baby Cure study was 95%. Notably, Baby Cure viruses were neutralized with a significantly greater potency (0.04 µg/ml) compared to FRESH viruses (0.17 µg/ml) (p = 0.02, Mann Whitney test) (Figs 3A- and 4). There was no difference in 3BNC117 neutralization coverage between FRESH and Baby Cure TF viruses, however neutralization potency was higher in infants (Geo mean $IC_{50}$ titers 0.19 µg/ml) compared to FRESH viruses (Geo mean $IC_{50}$ titers 0.46 µg/ml). Resistance to VRC01 was slightly higher in FRESH compared to Baby Cure TF viruses, with VRC01 only neutralizing 65% of viruses in the FRESH cohort and 76% in Baby Cure TF viruses, and with relatively high potency (Geometric mean $IC_{50}$ titers 0.96 and 0.71 µg/ml) for FRESH and Baby Cure respectively. Collectively, Baby Cure TF viruses were slightly more sensitive to the CD4 binding site bnAbs compared to FRESH viruses.

Interestingly, both V2-apex bnAbs CAP256-VRC26.25 and PGDM1400 had high neutralization coverage (74% and 76%) with a strong potency (Geo mean $IC_{50} < 0.10$ µg/ml) against FRESH TF viruses (Figs 3A and 4). In contrast, a large number of infants viruses were resistant to CAP256-VRC26.25 and PGDM1400 (57% and 67% coverage; Geometric mean < 0.1µg/ml) compared to FRESH viruses. However, infant viruses had a slightly higher neutralization coverage with lower potency to V3-glycan supersite mAbs including PGT121 and 10–1074 (71 and 76% coverage; Geometric mean $IC_{50} < 0.2$ µg/ml) (Fig 3C). In contrast, FRESH viruses were less sensitive to both PGT121 and 10–1074, with neutralization coverage at 57 and 59% respectively, but the potency (0.1µg/ml) was slightly higher than Baby Cure viruses.

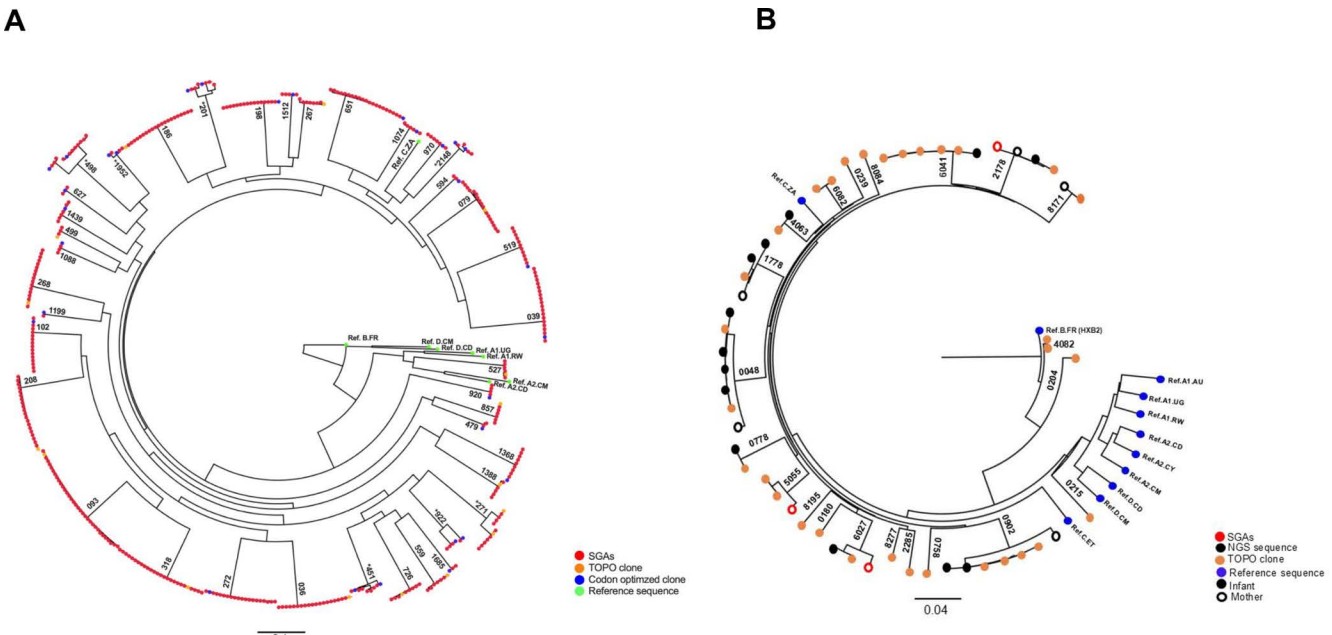

**Fig 2. Phylogenetic analysis of full-length HIV-1 Env transmitted/founder sequences generated from (A) FRESH participants (n = 39) and (B) Baby Cure infants (n = 19).** A maximum likelihood phylogenetic tree was constructed using 383 single genome sequences and a total of 46 *env* clones generated from representative single genome sequences. SGA sequenced are shown as red circles, TOPO clones shown as orange circles and codon-optimised clones are shown green circles. (*) denotes participants infected with more than one variant at the time of infection. All HIV-1 reference sequences (blue circles) representing genetic subtypes were obtained from the Los Alamos National Laboratory HIV Sequence Database included: Ref. A1.RW, Ref. A1.UG, Ref. A2.CM, Ref. A2.CD, Ref. D.CD, Ref. D.CM, Ref. B.FR and Ref. C.ZA.

Baby Cure viruses were slightly more sensitive to the MPER-bnAb 10E8 (100% coverage, geometric mean IC$_{50}$ 0.42 µg/ml) compared to FRESH viruses (89% coverage, geometric mean IC$_{50}$ 0.83) (p = 0.07) (Figs 3A, 3B and 4). Interestingly, we found a higher neutralization coverage (83% coverage) of FRESH viruses to gp120-gp41 interface PGT151 compared to Baby Cure viruses (57% coverage). However, no differences were observed in neutralization potency between FRESH and Baby Cure viruses (geometric mean IC$_{50}$ 0.09 µg/ml and 0.06 µg/ml respectively) (Fig 3C). Taken together, these findings suggest that there is heterogeneity among infants and adults TF viruses, particularly in their Env structure and N-linked glycosylation patterns that may lead to the differences in accessibility to the CD4 binding site, V2 apex and V3-glycan bnAbs neutralization sensitivity (breadth and potency). These distinct differences in the sensitivity of infant and adult transmitted/founder viruses may lead to potential differences in clinical application of these bnAbs in infants versus adults. The most effective bnAbs for adult subtype C TF viruses were VRC07–523LS (CD4bs), CAP256-VRC26.25 or PDGM1400 (V2-glycan), 10–1074 (V3 glycan supersite) and PGT151 (gp120-gp41 interface) (Fig 3A). In contrast, the most effective bnAbs for infant subtype C TF viruses were VRC07–523LS or 3BNC117 (CD4bs), 10–1074 or PGT121 (V3-glycan supersite) and 10E8 (MPER) (Fig 3B).

## Predicted neutralization coverage of triple combination of bnAbs

We used the Bliss-Hill model to predict the best combination of at least three (triple combination) bnAbs for the FRESH and Baby Cure cohorts. We analyzed the neutralization coverage of viruses by at least one active, two active and three active bnAbs as previously described [25]. In the FRESH cohort, we found multiple options of triple combination of bnAbs that achieved neutralization coverage of >90% with a single active bnAb (Fig 5A–C). The best triple combinations against

**A**

| | PSV | Subtype | CD4-binding site VRC01 | VRC07-523 | 3BNC117 | V1V2-Loop CAP256-VR | PGDM1400 | V3-Loop 10-1074 | PGT121 | MPER 10E8 | Gp120-gp41 PGT151 |
|---|---|---|---|---|---|---|---|---|---|---|---|
| 1 | 36 | C | 0,23 | 0,01 | 0,07 | 5,48 | 0,01 | 0,01 | 0,01 | 0,91 | 0,01 |
| 2 | 039 | C | 1,80 | 0,80 | 1,82 | >10 | >10 | >10 | >10 | 3,65 | 0,24 |
| 3 | 79 | C | 2,18 | 0,02 | >10 | 0,01 | 0,01 | >10 | >10 | 1,40 | 0,02 |
| 4 | 93 | C | 0,59 | 0,02 | >10 | 0,01 | 0,09 | 0,01 | 0,01 | 0,64 | 0,06 |
| 5 | 102 | C | >10 | 0,14 | 0,07 | 0,01 | 0,01 | >10 | 0,48 | 1,20 | 0,05 |
| 6 | 186 | C | 0,40 | 0,01 | 0,10 | 0,01 | 0,01 | 0,14 | 0,02 | 0,50 | 0,01 |
| 7 | 198 | C | >10 | 1,55 | >10 | 0,01 | >10 | 0,18 | 0,50 | 5,79 | 0,02 |
| 8 | 201A | C | >10 | 0,18 | 0,13 | 0,06 | 0,57 | 0,02 | 0,54 | 0,01 | >10 |
| 9 | 201B | C | >10 | >10 | >10 | 0,02 | 0,27 | 0,09 | 0,06 | 0,60 | 0,09 |
| 10 | 208 | C | >10 | 0,13 | 0,06 | 0,46 | 0,30 | 0,23 | 0,04 | 1,31 | 0,02 |
| 11 | 267 | C | 1,33 | 0,03 | 0,14 | 0,01 | >10 | >10 | 0,09 | 0,17 | 1,52 |
| 12 | 268 | C | >10 | >10 | >10 | >10 | 9,29 | 0,16 | 0,06 | 0,51 | 0,03 |
| 13 | 271A | C | 0,32 | 0,01 | 0,55 | 0,19 | >10 | >10 | >10 | 1,05 | 0,06 |
| 14 | 271B | C | 6,37 | 0,01 | >10 | 0,01 | 0,01 | >10 | >10 | 2,43 | 0,04 |
| 15 | 272 | C | >10 | 0,09 | >10 | 0,01 | 0,01 | 0,05 | 0,04 | 0,32 | 0,48 |
| 16 | 318 | C | 0,92 | 0,05 | 0,06 | 0,01 | >10 | 0,01 | 0,02 | 0,52 | 0,20 |
| 17 | 451A | C | >10 | 3,92 | 1,42 | >10 | >10 | 0,02 | 0,01 | 3,30 | 5,10 |
| 18 | 451B | C | >10 | 8,39 | 5,47 | >10 | >10 | 0,02 | 0,02 | 4,00 | 1,74 |
| 19 | 479 | C | 5,01 | 0,02 | 0,17 | 0,01 | 0,01 | >10 | >10 | 3,39 | 0,02 |
| 20 | 498A | C | >10 | 4,11 | >10 | 0,24 | 1,45 | >10 | >10 | >10 | 0,03 |
| 21 | 498B | C | >10 | 3,97 | 8,55 | 0,48 | 1,82 | >10 | >10 | >10 | 0,06 |
| 22 | 499 | C | 3,30 | 0,05 | 0,67 | 0,95 | 0,81 | >10 | >10 | 0,96 | 0,01 |
| 23 | 519 | C | >10 | 0,28 | >10 | >10 | 0,02 | >10 | >10 | 0,25 | 0,18 |
| 24 | 527 | AC | 0,14 | 0,11 | 0,26 | 0,01 | 0,01 | >10 | >10 | 0,09 | >10 |
| 25 | 559 | C | 0,55 | 0,01 | >10 | 0,67 | 0,12 | >10 | >10 | 0,17 | >10 |
| 26 | 594 | C | 5,47 | 1,46 | 6,42 | 1,56 | 6,36 | 0,09 | 0,14 | 1,33 | 0,05 |
| 27 | 627 | C | 1,71 | 0,40 | 0,34 | >10 | 1,69 | >10 | >10 | 0,38 | 0,07 |
| 28 | 651 | C | 0,43 | 0,13 | 0,13 | 0,12 | 0,03 | 0,43 | 1,55 | 6,13 | 0,08 |
| 29 | 726 | C | 0,30 | 0,22 | 0,35 | 0,01 | >10 | >10 | >10 | 0,09 | 0,50 |
| 30 | 857 | C | 0,54 | 0,22 | 0,07 | 0,01 | 0,04 | 0,21 | 0,01 | 1,32 | 0,05 |
| 31 | 920 | C | 1,26 | 0,13 | 0,21 | 0,40 | 0,17 | >10 | >10 | 3,76 | 0,28 |
| 32 | 922A | C | 7,72 | 3,79 | 1,84 | 0,01 | >10 | 5,80 | >10 | 2,65 | 0,09 |
| 33 | 922B | C | >10 | 1,10 | >10 | >10 | >10 | >10 | >10 | 2,29 | 0,01 |
| 34 | 970 | C | 0,20 | 0,02 | 0,25 | >10 | >10 | 0,02 | 0,01 | 0,05 | 0,05 |
| 35 | 1074 | C | 4,54 | 0,47 | 2,53 | >10 | 1,92 | 0,33 | >10 | 0,32 | 0,08 |
| 36 | 1088 | C | 1,66 | 0,37 | 1,19 | 0,01 | 0,04 | 0,77 | 4,97 | 1,33 | 0,26 |
| 37 | 1199 | C | >10 | >10 | >10 | 0,07 | >10 | 0,97 | 0,67 | >10 | >10 |
| 38 | 1368 | C | >10 | 0,25 | 1,14 | >10 | 7,59 | 0,64 | >10 | >10 | 0,02 |
| 39 | 1388 | C | 0,90 | 0,01 | >10 | 0,01 | 1,14 | 0,02 | 0,02 | >10 | 0,03 |
| 40 | 1439 | C | 0,14 | 0,03 | >10 | 1,72 | 0,02 | 1,25 | 4,81 | 0,38 | 0,07 |
| 41 | 1512 | C | >10 | 0,95 | 4,40 | 0,09 | 0,02 | >10 | >10 | 3,56 | 0,02 |
| 42 | 1685 | C | >10 | 8,14 | 6,15 | >10 | >10 | 0,04 | 3,61 | 3,72 | 0,02 |
| 43 | 1952A | C | 1,10 | 0,15 | 0,31 | 0,01 | 0,01 | 0,02 | 0,02 | 0,93 | 7,40 |
| 44 | 1952B | C | 1,36 | 0,16 | 0,40 | 0,01 | 0,01 | 0,05 | 0,04 | 3,53 | >10 |
| 45 | 2148A | C | 9,63 | 0,27 | 0,23 | >10 | >10 | 0,41 | 0,76 | 4,02 | 0,45 |
| 46 | 2148B | C | 4,98 | 0,64 | 1,30 | >10 | 0,02 | 0,22 | 0,16 | 1,27 | 0,24 |
| Geo Mean | | | 1,01 | 0,17 | 0,49 | 0,05 | 0,10 | 0,09 | 0,09 | 0,85 | 0,09 |
| IC50 Breadth | | | 65 | 93 | 70 | 74 | 76 | 59 | 57 | 89 | 83 |

**B**

| | PID's | Clade | CD4-binding site VRC01 | VRC07-523LS | 3BNC117 | V2 Apex CAP256-VR | PGDM1400 | V3 Glycan 10-1074 | PGT121 | MPER 10E8 | Gp120-gp41 PGT151 |
|---|---|---|---|---|---|---|---|---|---|---|---|
| 1 | 6041_1 | C | 0,08 | 0,01 | 0,08 | 0,005 | 0,01 | 0,01 | 0,01 | 0,52 | >10 |
| 2 | 0204_1 | BC | 0,06 | 0,01 | 0,01 | >10 | 5,46 | >10 | >10 | 0,01 | 0,24 |
| 3 | 6082_1 | C | 0,26 | 0,01 | 0,49 | 0,005 | 0,02 | 0,04 | 5,50 | 0,28 | >10 |
| 4 | 4082_1 | BC | 0,08 | 0,01 | 0,03 | >10 | >10 | >10 | >10 | 0,01 | 0,43 |
| 5 | 8195_1 | C | 0,11 | 0,01 | 0,01 | >10 | >10 | 0,005 | 0,60 | 1,56 | 0,01 |
| 6 | 0215_1 | AC | 0,15 | 0,06 | 0,04 | 0,005 | 0,005 | 1,95 | 2,94 | 0,02 | 0,03 |
| 7 | 0778_1 | C | 0,22 | 0,06 | 0,11 | 0,01 | 0,22 | 0,07 | 0,60 | 1,56 | 0,01 |
| 8 | 8171_1 | C | 0,81 | 0,03 | 0,19 | 0,24 | 0,32 | 0,61 | >10 | 1,65 | 0,06 |
| 9 | 8277_1 | C | 1,05 | 0,03 | 0,26 | >10 | 0,04 | 0,01 | 0,005 | 2,28 | >10 |
| 10 | 0239_1 | C | 1,4 | 0,04 | 0,89 | 0,01 | >10 | 0,13 | 0,12 | 0,47 | >10 |
| 11 | 2178_1 | C | 2,7 | 0,04 | 0,54 | 0,005 | >10 | >10 | >10 | 6,24 | >10 |
| 12 | 0902_1 | C | >10 | 0,04 | >10 | 0,005 | 0,005 | 0,06 | 1,05 | 3,94 | >10 |
| 13 | 1778_1 | C | 1,79 | 0,04 | 0,76 | 0,005 | 0,005 | 0,05 | 0,02 | 0,26 | 0,01 |
| 14 | 0048_1 | C | 4,9 | 0,09 | >10 | 0,24 | 3,3 | 0,01 | 0,05 | 0,07 | 0,03 |
| 15 | 5055_1 | C | >10 | 0,01 | >10 | >10 | >10 | 0,35 | 0,07 | 0,34 | 0,04 |
| 16 | 0180_1 | C | >10 | 0,19 | 0,71 | 0,005 | 0,005 | 0,05 | 0,05 | 0,13 | 1,18 |
| 17 | 4063_1 | C | 7,14 | 0,34 | 1,32 | >10 | 5,21 | >10 | >10 | 0,34 | >10 |
| 18 | 6027_1 | C | 9,64 | 0,54 | 1,26 | >10 | 3,31 | >10 | >10 | 0,35 | 0,21 |
| 19 | 0758_1 | C | >10 | 0,36 | >10 | >10 | >10 | 4,56 | 0,34 | 7,52 | 0,21 |
| 20 | 8084_1 | C | 5,66 | 0,15 | >10 | 0,01 | 0,01 | 0,02 | 1,36 | 0,67 | 0,01 |
| 21 | 2285_1 | C | >10 | >10 | >10 | >10 | >10 | 0,05 | 0,04 | 2,52 | 0,12 |
| Geo mean | | | 0,71 | 0,04 | 0,19 | 0,01 | 0,08 | 0,07 | 0,18 | 0,42 | 0,06 |
| IC50 Breath | | | 76 | 95 | 71 | 57 | 67 | 76 | 71 | 100 | 57 |

IC50 titers µg/ml

| | |
|---|---|
| | <0,1 |
| | 0,1-0,9 |
| | 1,0-3,0 |
| | 3,1-9,9 |
| | >10 |

**Fig 3. Neutralization sensitivity of the transmitted/ founder viruses to nine broadly neutralizing antibodies that target the CD4 binding site (3BNC117, VRC01, VRC07-523LS), the V1V2 loop (CAP256-VRC26.25, PGDM1400), V3-supersite (PGT121, 10-1074), MPER (10E8) and gp120-gp41 interface (PGT151) in the FRESH cohort (A) and Baby Cure study (B).** The IC$_{50}$ titers are indicated as percentages. Red coloured squares indicate neutralization IC50 scores <0,1, orange coloured squares indicate neutralization IC$_{50}$ between 0,1- 0,9, Yellow squares indicate IC$_{50}$ titers between 1,0-3,0 and light yellow colour indicates IC$_{50}$ titers between 3,1-10µg/ml. Clear squares indicate IC$_{50}$ titers>10. The tier phenotyping for neutralization sensitivity or resistance of the viruses tested are not available.

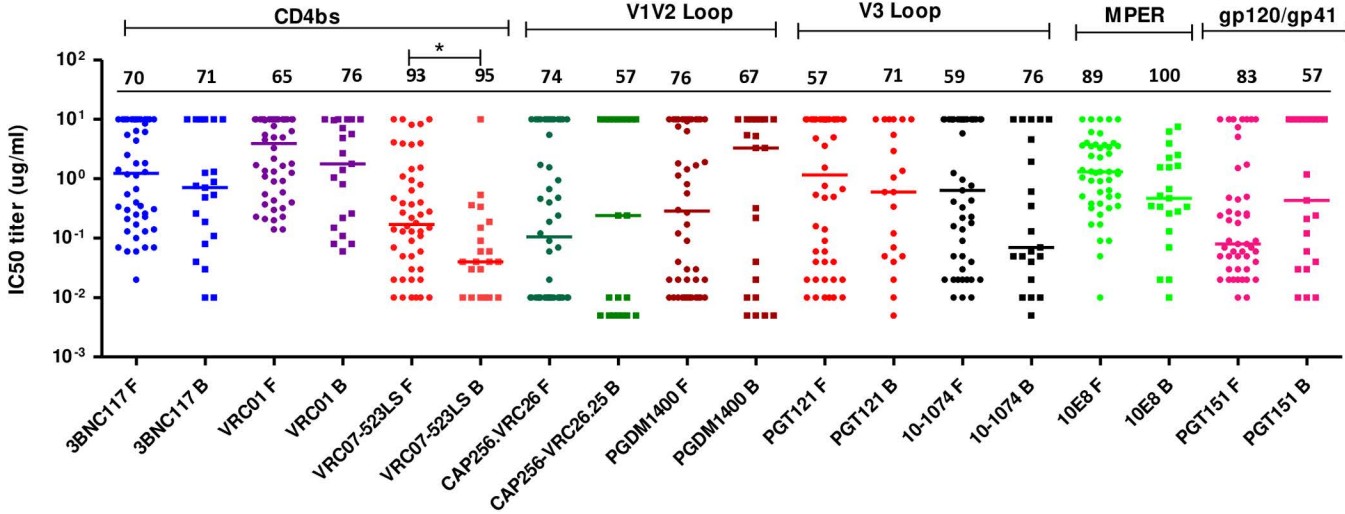

**Fig 4. Neutralization coverage of transmitted/founder viruses produced from the FRESH cohort indicated by F and Baby Cure as indicated by B.** Each virus is represented by an individual dot, the highest concentration of each antibody tested against all the pseudoviruses was 10µg/ml. Solid lines represent the median titers. Percentage IC$_{50}$ neutralization coverage is shown above each scatter plot. Significant differences were observed in VRC07-523LS and 10E8, statistical significance was determined using Mann Whitney *t*-test, which does not correct for multiple comparisons. * indicates p < 0.05.

the FRESH subtype C viruses was CAP256.VRC26.25/PGT151/VRC07–523LS; 10–1074/PGT151/VRC07–523LS or PGT121/ PGT151/VRC07–523LS and all achieved 93% coverage, with a single active bnAb (Fig 5A–C). However, neutralization coverage was less than 50% in viruses that were sensitive to at least two bnAbs (dual active) and <10% coverage was observed against viruses neutralized by all three bnAbs (three active). Notably, both CAP256.VRC26.25/PGT151/VRC07–523LS and 10–1074/PGT151/VRC07–523LS combinations achieved 46% coverage for dual active bnAbs and they were the best combinations for FRESH TF viruses.

In the Baby Cure cohort, we also found multiple triple combinations that achieved >90% neutralization coverage with a single active bnAb; VRC07–523LS/PGT121/10E8 achieved 100% neutralization coverage when considering "one active" bnAb (Fig 5D). While, VRC07–253LS/10–1074/10E8/ and VRC07–523LS/10E8/PGT151 both achieved 95% neutralization coverage on a single active bnAb (Fig 5E and 5F). In addition, VRC07–523LS/10–1074/10E8 had the highest "two active" coverage of 62% compared to 43% for VRC07–523LS/10E8/PGT121 and 10E8/VRC07–523LS/PGT151. These findings suggest that VRC07–523LS/10–1074/10E8 was the best combination for the Baby Cure TF viruses.

### Higher frequency of CD4 binding site resistance mutations in adults compared to infant sequences

We next sought to determine the differences in viral Env amino acid signatures between the neutralization sensitive and resistant viruses. We analysed specific motifs of the HIV-1 *env* clonal sequences from both the FRESH and Baby Cure cohorts and evaluated their association with neutralization sensitivity to bnAbs. We first analysed amino acid residues in the loop D, the N-linked glycans were highly conserved at position 276 in both FRESH and Baby Cure viruses. However, half of FRESH viruses had an asparagine or serine (279N/S) mutations and they were partially resistant to CD4bs bnAbs (Fig 6A and 6B). Interestingly, we also found a rare glutamic acid mutation in at least two FRESH and one Baby Cure TF sequences at position 279 (N279E) and these clones were completely resistant to all CD4bs bnAbs. Furthermore, asparagine was highly conserved at position 280, although a small fraction of FRESH participants had serine substitution and they were resistant to CD4 binding site bnAbs (Fig 6A and 6B).

We also analysed the CD4 binding loop, serine (S) residues were highly conserved at position 365 in both FRESH and Baby Cure sequences, although threonine (T) and proline (P) mutations were observed in 6% of the FRESH participants. In contrast, 24% of Baby Cure sequences had alanine (A) residues, which was found in both sensitive and resistant clones. In addition, at position 368, aspartic acid (D) was highly conserved in both FRESH and Baby Cure sequences and there were no escape mutations observed (Fig 6A and 6B). Furthermore, we analysed the β20-β24 in the base of the V5 loop, and an arginine (R) at position 456 was found in sensitive viruses, while tryptophan (W), histidine (H) and serine (S) (456W/H/S) mutations were found in partially resistant viruses. In addition, glycine (G) residues at position 458 and 459 were highly conserved in both cohorts and these were previously associated with sensitivity to CD4 binding site antibodies [60,61]. While, one clone (268) in the FRESH cohort had a lysine (G458K) mutation, another clone (1199) had an aspartic acid (G459D) mutation, and these clones were completely resistant to all three CD4bs mAbs.

### High frequency of escape mutations in the V2-Apex are associated with V2-apex bNAb resistance in infant TF clones

We first analysed the N-linked glycans at position 156 and 160; in the FRESH cohort, both N156 and N160 glycans were highly conserved in most of the *env* sequences, these glycans were previously associated with high sensitivity to PGDM1400 mAb and CAP256-VRC26.25 (Fig 6C and 6D). Interestingly, single point mutations were observed at position 156 involving arginine, lysine, tyrosine, aspartic acid and asparagine (N156R/K/Y/D/N). Furthermore, additional mutations were observed at position 160 involving lysine, serine, tyrosine, aspartic acid and glutamic acid (N160K/S/E/Y/D), these *envs* were resistant to both PGDM1400 and CAP256-VRC26.25. In contrast, only one (2285–1) Baby Cure sequence had arginine (N156R) mutation, while lysine, serine and tyrosine mutations were found in position 160 (N160K/S/Y), these clones were resistant to these V2-apex bnAbs. (Fig 6C and 6D).

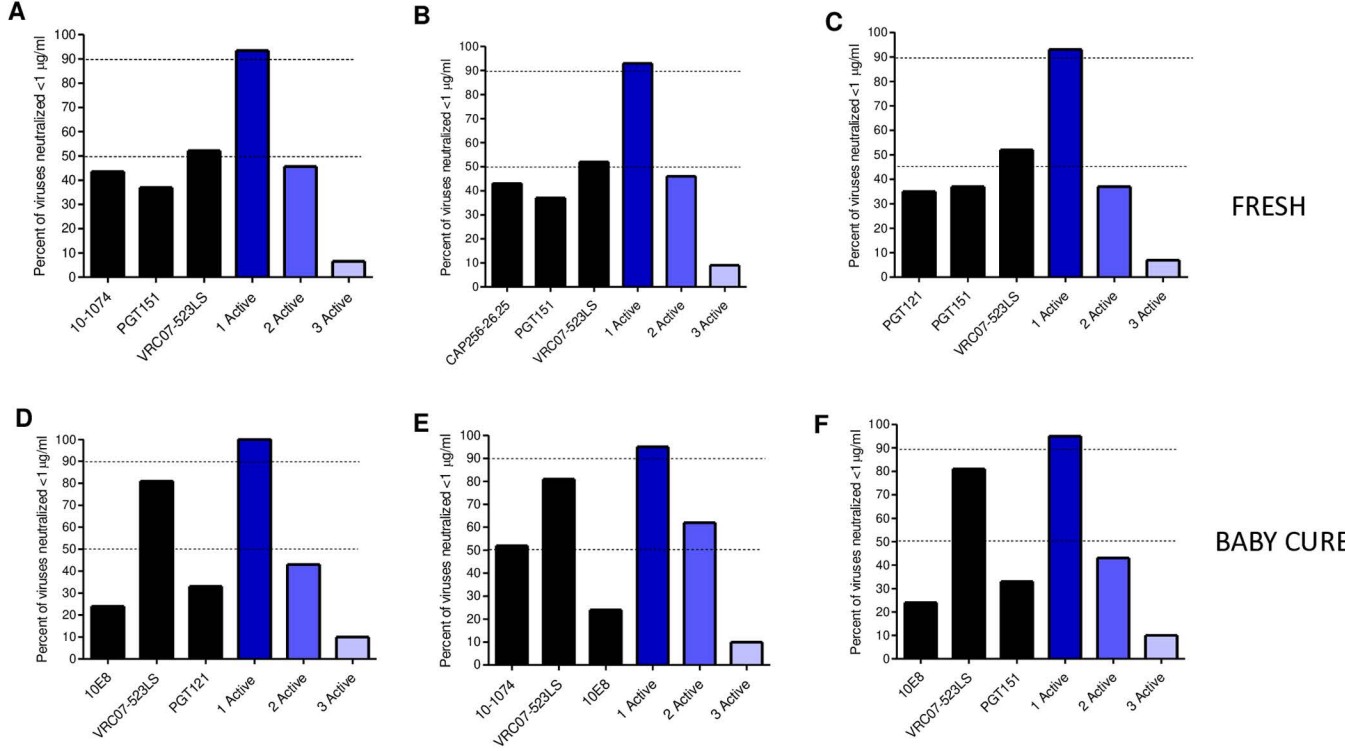

**Fig 5. Predicted neutralization coverage of triple bnAb combinations against FRESH (A-C) and BABY CURE (D-F) transmitted/ founder viruses (IC80 < 1 µg/ml).** Neutralization coverage was predicted using the Bliss Hill model, neutralization coverage on single bnAbs is shown in black, while neutralization on triple combinations is shown in different shades of blue. Neutralization coverage on single "active" bnAb is shown in dark blue, two "active" bnAbs is shown in blue and three "active" bnAbs is shown in light blue. The dotted lines show neutralization coverage at 50% and 90%.

Furthermore, majority of FRESH and Baby Cure TF sequences had a lysine at position 169 (K169), while the remaining sequences had glutamine, glutamic acid, threonine, arginine, serine, asparagine, valine and isoleucine (K169E/Q/R/S/I) and they were resistant to PGDM1400 and CAP256-VRC26.25. Interestingly, the frequency of substitutions at position 169 was higher (40%) in infants TF compared to 28% in adults and this could explain the increased resistance of infant TF clones to CAP256-VRC26.25 mAb. We also analysed amino acid signatures at positions 164, 166 and 167 as these are important contact sites for V2-apex bnAbs [62]. Glutamic acid was highly conserved at position 164 (E164) in both infant and adult clones and it was observed in CAP256-VRC26–25 sensitive variants. However, serine, glycine and valine (E164S/G/V) substitutions were observed in clones that were completely resistant to both PGDM1400 and CAP256-VRC26.25. Arginine was highly conserved at position 166 and was found in sensitive viruses, while few clones had lysine and threonine mutations (R166K/T) and they were resistant to CAP256-VRC26.25 only. Few infant TF viruses also had a glycine (R166G) mutation and they were also resistant to CAP256-VRC26.25. Furthermore, aspartic acid was highly conserved at position 167 in both adults and infants, however two adult TF clones had asparagine (D167N) mutations and both were resistant to CAP256-VRC26.25. We therefore identified several mutations that may be associated with resistance to the V2-apex directed mAbs, the impact of unusual mutations in neutralization sensitivity will require validation through site-directed mutagenesis studies.

## High frequency of V3-glycan supersite substitutions in adults than infants T/F clones

More than 90% of the Env clones that were sensitive to both 10–1074 and PGT121 had an N-linked glycan at position 332 confirming the importance of this glycan to most V3-loop bnAbs (Fig 6E and 6F). While, Env clones that were resistant to

**FRESH**  **BABY CURE**

**A: CD4 binding site**

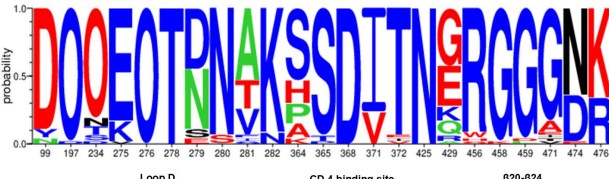

**B: CD4 binding site**

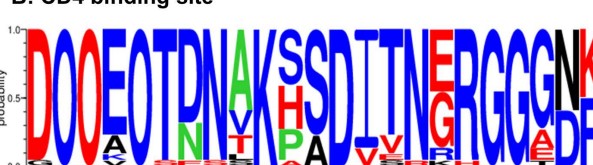

**C: V2-glycan**

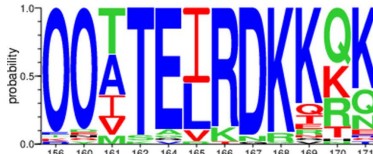

**D: V2-glycan**

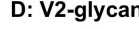
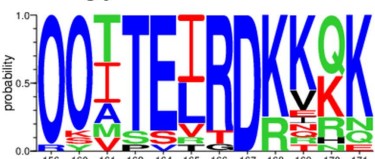

**E: V3-glycan**

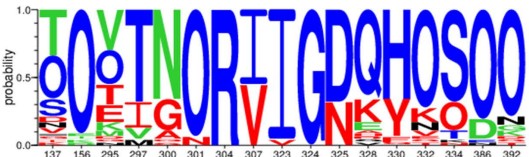

**F: V3-glycan**

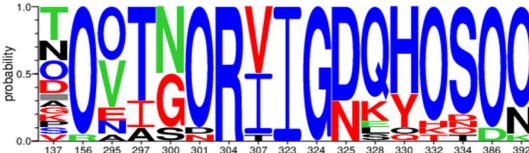

**G: MPER**

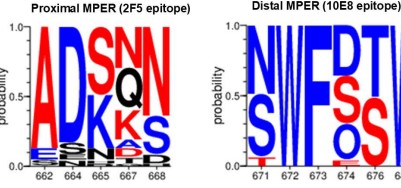

**H: MPER**

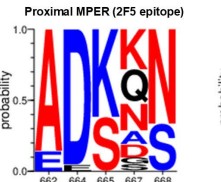
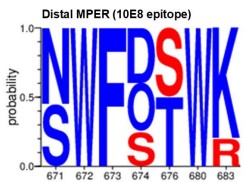

**I: Gp 120-gp41 interface**

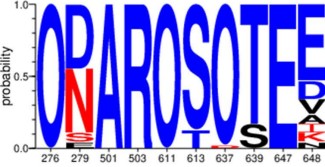

**J: Gp 120-gp41 interface**

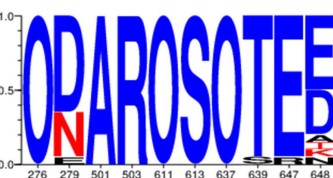

**Resistant**
**Sensitive**
**Intermediate/undetermined**
**Uncharacterized**

**Fig 6. Sequence logo plots T/F viruses from FRESH and Baby Cure studies showing the important amino acid positions within the neutralizing epitopes targeting the CD4 binding site (A, B), V2-glycan (C, D); V3-glycan (E, F), proximal (G, H) and distal MPER (I, J) regions and Gp120-gp41 interface (K, L).** The size of each AA in the logogram is proportional to its frequency and the "O" represents N-linked glycans. The numbering of each AA corresponds to the residue positions of HXB2. Residues associated with a sensitive neutralization phenotype are in blue, residues in red represent resistant residues while green represents differences in AAs between antibodies of the same class.

both 10–1074 and PGT121 had lysine, valine, serine, glutamic acid, arginine and threonine (N332K/V/S/E/R/T). Interestingly, infants had a higher frequency of the canonical N332-glycan compared to adults, which could explain the differences in neutralization coverage between infants and adults. The N-linked glycan at position 301 was highly conserved in both infant and adult TF clones. However, one of the infant clones had an aspartic acid (D) at position 301 and this TF clone was resistant to both 10–1074 and PGT121. We also analysed the N-linked glycans at position 295, the clones that were resistant to V3 loop bnAbs had threonine, glutamic acid, lysine, aspartic acid, glycine, methionine or alanine at position 295 (295T/E/K/D/G/M/A). In addition, we analysed the GDIR sequence (HXB2 positions 324–327), 30% of TF clones had an asparagine substitution at position 325 (D325N) in both infants and adult TF which was previously associated with resistance to V3-loop bnAbs [63]. Finally, we analysed the V1 loop length and the number of N-linked glycosylation sites (PNGS), there were no significant differences in the V1 loop length and the number of PNGS between infants and adult TF sequences (Fig 7). However, few clones (11%) in FRESH (1368, 920, 627, 499, 208) and Baby Cure (24%) cohorts (0215–1, 0758–1, 2285–1, 6027–1, 0239–1) had longer V1 loops (37–43 AA) and they were either partially or completely resistant to V3-glycan supersite bnAbs. Together, these findings highlight a high frequency of V3-loop mutations in adult compared to infant TF clones, while the frequency of V1 loop insertions was slightly higher in infants compared to adults and this may explain the minor differences in neutralization sensitivity to V3-glycan supersite bnAbs.

### Higher frequency of escape mutations in the proximal MPER, while the distal MPER and gp120-gp41 interface epitopes were highly conserved

We first analysed the proximal MPER region (HXB2 position 662–667), a target epitope for 2F5 mAb (Fig 6G and 6H) [64]. The frequency of glutamic acid (E) at position 662 was extremely low, while the frequency of alanine mutation was high and it is associated with 2F5 resistance. Aspartic acid (D) residues were highly conserved at position 664, with minor variations in both cohorts. At position 665, lysine (K) residues were only found in half of the infants and FRESH variants, a high frequency of resistant mutations was observed including serine, asparagine, glutamic acid and they are associated with 2F5 resistance. Notably, alanine at position 667 is associated with sensitivity to 2F5 antibody [61]; although the frequency of alanine residues was extremely low in both FRESH and Baby Cure variants. Similarly, previous studies have shown that subtype C viruses have high 2F5 resistance due to the absence of alanine at position 667 [61].

We also analysed amino acid signatures within the distal MPER, which contains 10E8/4E10 contact positions of the membrane proximal external region (MPER) of gp41 (HXB2 position 671–683) [65]. The distal MPER region was highly conserved (Fig 6I and 6J), most of the sequences had either asparagine or serine (N/S) at position 671 and both residues are associated with sensitivity to the 10E8 mAb, while less than 10% of Env clones in adults had a threonine (T) that is associated with resistance to the 10E8 mAb. Tryptophan (W) and phenylalanine (F) at positions 672 and 673 respectively were found in 100% of both adults and infant sequences and both residues are associated with sensitivity to 10E8 [61].

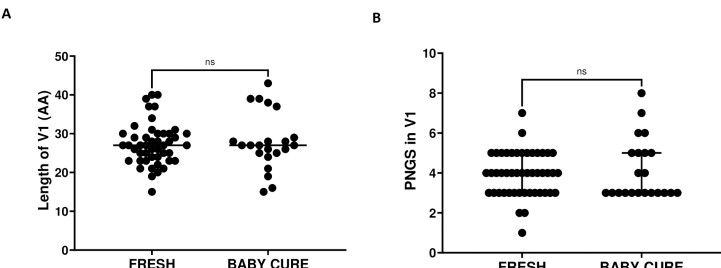

**Fig 7. Amino acid length of the V1 Loop and Potential N-linked glycosylation sites (PNGS) in the V1 Loop on HIV-1 TF variants isolated from FRESH and Baby Cure cohorts in the first week or one month of infection.**

Moreover, a low frequency of sequences had serine (S), aspartic acid (D) or asparagine (N) at position 674, however they were found in both 10E8 mAb sensitive and resistant clones. These findings indicate a conserved distal MPER epitope, while the proximal MPER had a high frequency of resistant mutations in both adults and infants, suggesting that bnAbs targeting the distal MPER epitope, including 10E8, 4E10 and DH511 may be effective vaccine candidates in subtype C settings.

Finally, we also analysed amino acid residues on the gp120-gp41 interface that is an important epitope for the PGT151 mAb (Fig 6K and 6L). The N-linked glycans at position 611 and 637 are crucial for PGT151 neutralization [66], and these glycans were highly conserved in both adult and infant TF sequences except for one adult clone (559) that had a D substitution and this clone was resistant to the PGT151 bnAb (Fig 6K and 6L). Our findings suggest that gp120-gp41 interface epitope could be targeted for heterosexual but not vertical transmission of subtype C variants.

## Discussion

This study characterized the sensitivity of transmitted/founder viruses from the FRESH and Baby Cure cohorts to bnAbs targeting different epitopes on the HIV-1 Env glycoprotein. The viruses were transmitted through heterosexual or *in-utero* routes respectively. Our findings suggest that the combination of VRC07–523LS, CAP256-VRC26 or 10–1074 and PGT151 mAbs are highly effective against heterosexual subtype C infections. Moreover, the combination of VRC07–523LS, 10E8 and 10–1074 or PGT121 may be more effective in preventing vertical transmission of HIV-1. Interestingly, high resistance in Baby Cure T/F clones was noted for CAP256-VRC26, PGDM1400 and PGT151, while slightly higher resistance to PGT121 and 10–1074 was observed for the FRESH study clones.

Analysis of the D-loop, CD4 binding loop, V5 loop and the bridging sheet revealed a low frequency of escape mutations within the CD4bs in both cohorts. We noted a higher frequency of V2 apex escape mutations in the Baby Cure compared to FRESH participants and these mutations were associated with resistance to V2 apex bnAbs. In addition, some of the FRESH TF clones were missing the N-linked glycan at position 332, and this was associated with resistance to V3-glycan bnAbs. Finally, the proximal MPER region had frequent escape mutations, while the distal MPER region and gp120-gp41 interface were highly conserved in both FRESH and Baby cure clones, despite the differences observed in PGT151 neutralization sensitivity.

Previous studies characterized subtype C viruses and demonstrated that VRC07–523LS, CAP256-VRC26.25 and 10E8 were the most broad and potent bnAbs against subtype C viruses [24,25]. Interestingly, mutations that were found in both FRESH and Baby Cure TF viruses are common escape mutations that were previously reported in other donors with subtype C infection [24,25,29,67]. Similarly, other studies reported a high resistance of subtype C viruses to V3-glycan supersite antibodies including PGT121 and 10–1074 [26,48] due to underrepresentation of the N332 glycan in subtype C viruses [28]. Together, these studies suggested that a combination of VRC07–523LS, CAP256-VRC26.25 and 10E8/PGT151/10–1074/PGT121 mAbs are effective combinations for prevention or treatment of subtype C infections.

Furthermore, previous studies characterized transmitting and non-transmitting mother-child pairs and provided insights into the correlates of vertical transmission and the potential mechanisms of viral transmission from mothers to infants [40,49,68–72]. The studies showed that the magnitude of maternal non-neutralizing and autologous neutralizing IgG responses were associated with a lower risk of vertical transmission [40,69,70]. However, transmitting mothers may preferentially transmit neutralization-resistant variants due to immune escape from maternal plasma bnAbs [49,69,72]. Overall, these studies demonstrated that the majority of transmitting mothers harbour both neutralization sensitive and resistant strains, however they may preferentially transmit V2 apex, V3 glycan and CD4-bs bnAb resistant strains to their infants [69]. Our findings add to this data and suggest a selection for V2-apex and gp120-gp41 interface escape mutants during in-utero vertical transmissions, although the underlying mechanisms remain unresolved.

Moreover, analysis of full-length subtype C *env* sequences in mother-infant pairs showed that in-utero transmissions selected for variants with shorter and less glycosylated V1 and V5 loops [71,72] and these features are associated with

increased sensitivity to V3 glycan and CD4 binding site bnAbs. However, transmission through breastfeeding selected for variants that had similar variable loops to their maternal variants, with fewer PNGS in gp41 and they were more sensitive to V2 glycan and MPER bnAbs [71]. We also observed a similar pattern, most of our in-utero infant *env* sequences had shorter V1 loops and were highly sensitive to the CD4 binding site, V3 glycan and MPER bnAbs. However, resistance to V2 apex and gp120-gp41 interface bnAbs is a great concern. Previously, another study reported transmission of similar variants with few N-linked glycosylation sites in mother-infant pairs, that were different from viruses transmitted through sexual transmission in discordant couples [72]. These findings suggest that the antigenic features of variants that are selected for vertical transmission may differ from sexual transmission and these distinct features lead to differences in their sensitivity to bnAbs. These findings highlight the importance of continuous surveillance and characterization of currently circulating TF viruses, which may differ between vertical and horizontal HIV-1 transmission, with potential implications for immunogen design strategies. Furthermore, these findings suggest the importance of using some of these recently transmitted *envs* with increased sensitivity to bnAbs to design vaccine immunogens for both paediatric and heterosexual transmissions. These immunogens could be used as boosting immunogens for germline targeting vaccines. However, further studies are required to further characterize vertical transmissions and elucidate mechanisms that lead to differences in neutralization sensitivity.

This is one of the few studies to report differences in neutralization sensitivity between adults and paediatric populations especially for transmitted/founder viruses. However, there are limitations to this study including the small sample size for both FRESH and Baby Cure cohorts due to limited sample availability. In FRESH, a median of six single genome amplicons were generated for each study participant, and intrapatient nucleotide diversity was extremely low in most of participants (<1%) suggesting highly homogenous viral populations although the small number of amplicons calls for some caution in interpretation. Another limitation of our study is that in the Baby Cure study, most *env* clones were produced via bulk PCR and they may not accurately represent the viral population that were circulating in the infants, although this is mitigated by sampling as close as possible to the time of birth. An additional limitation for our study is that maternal viral sequences were not comprehensively characterized in the Baby Cure cohort, which precludes an understanding of the mechanisms of vertical transmissions and the preferential selection of escape mutants for in-utero transmission. Moreover, this study only investigated the sensitivity of TF viruses to the neutralization activity of bnAbs, we did not determine the ability of bnAbs to mediate Fc-effector-driven functions such as antibody-mediated cellular cytotoxicity (ADCC), antibody-mediated cellular phagocytosis (ADCP) or complement deposition, which may also contribute to the functional efficacy of bnAbs.

In summary, this study suggests that the triple combination of VRC07–523LS, CAP256-VRC26.25 and 10–1074 or PGT151 may be effective in preventing HIV-1 subtype C heterosexual transmissions, whereas the combination of 3BNC117, 10–1074 and 10E8 may be preferable in preventing vertical transmissions. The results indicate that intervention studies may have to consider different antibody combination in adult versus paediatric settings. The findings also suggest high transmission of V2-apex and gp120-gp41 interface escape variants in *in-utero* vertical transmission. However, it is unclear why transmission of these escape variants is higher in vertical compared to heterosexual transmission. We have also identified amino acid signatures that may mediate resistance to various bnAbs, including within bnAbs of the same class. This information may be important in the selection of bnAbs that will undergo clinical testing, designing tools that predict neutralization sensitivity and immunogen design prioritization to elicit bnAbs most likely to be effective in clinical settings.

## Supporting information

**S1 Table. Intrapatient HIV-1 diversity scores among FRESH participants.** FRESH PID = Participant Identifier; Intrapatient diversity score calculated as the average pairwise nucleotide difference; % diversity = intrapatient diversity score expressed as a percentage.
(DOCX)

**S1 Fig. Highlighter plots showing intrapatient HIV-1 sequence diversity in FRESH participants.** Representative Highlighter plots were selected to illustrate typical levels of intrapatient diversity observed among FRESH participants. Each horizontal line represents a single viral sequence derived from HIV-1 *env* single genome amplification (SGA) for individual participants. Colored vertical ticks indicate the positions and types of nucleotide substitutions relative to the participant-specific consensus sequence and the representative *env* clone used for neutralisation assays. Substitution types are color-coded as follows: A→G (green), G→A (yellow), C→T (red), T→C (light blue), transversions (other colors). (TIFF)

**S2 Table. FRESH and Baby Cure Neutralization $IC_{50}$ and $IC_{80}$ titers against nine bnAbs that target the CD4 binding site, V1V2 loop, V3-supersite, MPER and gp120-gp41 interface.** Red coloured squares indicate neutralization $IC_{50}/IC_{80}$ scores <0,1, orange coloured squares indicate neutralization $IC_{50}/IC_{80}$ titers between 0,1–0,9. Yellow squares indicate $IC_{50}/IC_{80}$ titers between 1,0–3,0 and light yellow colour indicates $IC_{50}/IC_{80}$ titers between 3,1–10 μg/ml. Clear squares indicate titers>10. (XLSX)

## Acknowledgments

We would like to thank Qiniso Mkhize, Qiniso Mthethwa, Raveshni Durgiah, Dale Kitchin and Bronwen Lambson for technical support. Broadly neutralizing monoclonal antibodies were donated by Penny Moore from the National Institute for Communicable Diseases, Johannesburg; Michel Nussenzweig, The Rockefeller University, New York, USA; Nicole Doria-Rose, Vaccine Research Center, National Institutes of Health, Bethesda, USA. Raltegravir used for immediate treatment was donated by Merck & Co., Inc. The authors thank all participants in the FRESH, the Baby Cure study participants and their guardians, without whom this study would not have been possible.

## Author contributions

**Conceptualization:** Bongiwe Ndlovu, Kamini Gounder, Nelisiwe Zikhali, Jennifer Mabuka, Bruce D. Walker, Philip J.R. Goulder, Thumbi Ndung'u.

**Data curation:** Bongiwe Ndlovu, Kamini Gounder, Nelisiwe Zikhali, Lanish Singh, Ntokozo Ntshangase, Nombali Gumede, Jane Millar, Nicholas E. Grayson, David Bonsall, Sandra E. Chaudron, Jennifer Mabuka, Penny L. Moore.

**Formal analysis:** Bongiwe Ndlovu, Kamini Gounder, Nelisiwe Zikhali, Lanish Singh, Ntokozo Ntshangase, Nombali Gumede, Jane Millar, Rebecca T van Dorsten, Nicholas E. Grayson, David Bonsall, Sandra E. Chaudron, Jennifer Mabuka, Penny L. Moore, Philip J.R. Goulder.

**Funding acquisition:** Bongiwe Ndlovu, Bruce D Walker, Kamini Gounder, Thumbi Ndung'u.

**Investigation:** Bongiwe Ndlovu, Kamini Gounder, Nelisiwe Zikhali, Jane Millar, Thumbi Ndung'u.

**Methodology:** Bongiwe Ndlovu, Kamini Gounder, Nelisiwe Zikhali, Lanish Singh, Ntokozo Ntshangase, Nombali Gumede, Jane Millar, Nicholas E. Grayson, David Bonsall, Sandra E. Chaudron, Jennifer Mabuka, Philip J.R. Goulder.

**Project administration:** Bongiwe Ndlovu, Kamini Gounder, Nelisiwe Zikhali, Thumbi Ndung'u.

**Resources:** Krista L. Dong, Bruce D. Walker, Penny L. Moore, Thumbi Ndung'u.

**Supervision:** Penny L. Moore, Philip J.R. Goulder, Thumbi Ndung'u.

**Validation:** Rebecca T van Dorsten.

**Writing – original draft:** Bongiwe Ndlovu.

**Writing – review & editing:** Kamini Gounder, Nelisiwe Zikhali, Jane Millar, Nicholas E. Grayson, Sandra E. Chaudron, Krista L. Dong, Bruce D. Walker, Penny L. Moore, Philip J.R. Goulder, Thumbi Ndung'u.

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
