## [Decision Letter · Decision Letter 0]

Distinct neutralization sensitivity between adult and infant transmitted/ founder HIV-1 subtype C viruses to broadly neutralizing monoclonal antibodies

PLOS Pathogens

Dear Dr. Ndung’u,

Thank you for submitting your manuscript to PLOS Pathogens. After careful consideration, we feel that it has merit but does not fully meet PLOS Pathogens's publication criteria as it currently stands. Therefore, we invite you to submit a revised version of the manuscript that addresses the points raised during the review process.

Please submit your revised manuscript within 60 days May 09 2025 11:59PM. If you will need more time than this to complete your revisions, please reply to this message or contact the journal office at plospathogens@plos.org. Please include the following items when submitting your revised manuscript:

We look forward to receiving your revised manuscript.

Kind regards,

R. Keith Reeves

Academic Editor

PLOS Pathogens

Susan Ross

Section Editor

PLOS Pathogens

Editor-in-Chief

PLOS Pathogens

orcid.org/0000-0003-2946-9497

Editor-in-Chief

PLOS Pathogens

orcid.org/0000-0002-7699-2064

**Journal Requirements:**

At this stage, the following Authors/Authors require contributions: Sandra E. Chaudron. Please ensure that the full contributions of each author are acknowledged in the "Add/Edit/Remove Authors" section of our submission form.

https://journals.plos.org/plospathogens/s/submission-guidelines#loc-parts-of-a-submission

- TM on pages: 8, and 10.

5) Please upload a copy of Figures 2C , and 3C which you refer to in your text on pages 12, and 17. Or, if the figures are no longer to be included as part of the submission please remove all reference to them within the text.

6) We have noticed that you have uploaded Supporting Information files, but you have not included a list of legends. Please add a full list of legends for your Supporting Information files after the references list.

7) Thank you for stating "HIV-1 env nucleotide sequences are available in GenBank database. Accession numbers for the FRESH cohort sequences are ranging from PQ874248-PQ874674. In addition, GenBank accession numbers for the BABY CURE cohort are ranging from PV016540-PV016586." Please note that, though access restrictions are acceptable now, your entire minimal dataset will need to be made freely accessible if your manuscript is accepted for publication. This policy applies to all data except where public deposition would breach compliance with the protocol approved by your research ethics board. If you are unable to adhere to our open data policy, please kindly revise your statement to explain your reasoning and we will seek the editor's input on an exemption.

8) Please amend your detailed Financial Disclosure statement. This is published with the article. It must therefore be completed in full sentences and contain the exact wording you wish to be published.

9) Please ensure that the funders and grant numbers match between the Financial Disclosure field and the Funding Information tab in your submission form. Note that the funders must be provided in the same order in both places as well.

**Reviewers' Comments:**

Reviewer's Responses to Questions

**Part I - Summary**

Reviewer #1: This study by Bongiwe et al. nicely evaluated sensitivity to TF viruses obtained from adult and pediatric populations to known clinically-relevant bnAbs. The results of their study highlighted differences in bnAb sensitivities to these viruses with important considerations for future passive immunization strategies in pediatric individuals living with HIV. The limitations of the study were addressed to some extent, but this study reported important findings that will advance the field of neonatal HIV immunity.

Reviewer #2: This manuscript describes the neutralization sensitivity of TF viruses identified in 2 different groups: within seven days after first detection in young women between 18-23 (FRESH cohort) or within 1 month after birth from in-utero infected infants (baby cure cohort). Overall, this paper generated Env sequences and clones from the 2 cohort groups, and then tested 47 adult and 21 baby viruses against a panel of 9 bNAbs targeting different sites on the trimer. This paper generated a very useful dataset of not only neutralization susceptibility of circulating viruses but also matched genotypic variants associated with resistance or sensitivity to bNAbs. Previous studies using older viruses predicted a best triple bNAb combination but is out of date for contemporaneous circulating viruses. This study demonstrated that the young females had viruses that would be best targeted by a triple bNAb combination recently described for adults, based on Envs from the AMP trial results, but interestingly, the baby viruses would be best targeted by a different combination. It is quite striking that there is under 50% dual coverage based on IC80 <1 �g/ml in the susceptibility analysis. This dataset is not paradigm shifting, but it adds important sequence and phenotype data including viruses recently transmitted to infants.

Reviewer #3: This manuscript evaluates the neutralization breadth and potency of HIV bNAbs for viruses amplified from plasma of young women and children with new HIV infection. The authors describe differences in neutralization breadth and potencies of 9 bNAb (or combinations of bNAbs), as well as in the detection of mutations conferring bNAb resistance, between viruses derived from the women vs. those derived from the children. They conclude that clinical trials evaluating the efficacy of bNAbs for preventing or treating HIV may need to consider the use of different bNAbs in adults vs. children.

With the growing body of preclinical and early clinical studies demonstrating the efficacy of bNAbs in preventing infection or restricting rebound off ART, this work will likely be of interest to the field. The general approach is technically sound. However, a few concerns temper overall enthusiasm.

Comments:

1. Many published studies characterize viruses from repository samples, which may be years to decades old. The samples used in these studies are taken from ongoing cohorts but it is unclear as to whether the samples are truly “contemporaneous.” When were the samples used in these studies obtained?

2. For these studies, HIV was amplified from plasma obtained from a prospective cohort of women who were monitored twice weekly using nucleic acid testing for primary infection. Samples used in these studies were taken within 7 days (median1 day) following first detection of nucleic acids. Of note, almost half of the women were Fiebig 2-4 at testing and many of these women had extremely high viral loads. Was either PacBio or Illumina sequencing done on the FRESH samples? A median of only 6 SGA env were generated for study; it would be interesting to see how representative the SGA env used in the assays were of the viral population present at the time of sampling.

3. The majority (16) of infant samples used in these studies were taken within 48 hours of birth, while a few (4) were obtained at one month. It would be helpful in Table 1 to distinguish the timing of samples designated as obtained at birth, as well as the prophylactic ART (NVP +/- ZDV) that the infants received. SGA’s were not done due to limited plasma sample volumes and only a single env clone was generated and tested in most infants yet from Table 2, it looks like the viral loads in most samples were high enough to allow SGA using very small amounts of plasma.

4. As the authors point out, only a very few matched maternal-infant pairs were characterized in this study, which seems like a missed opportunity to examine mechanisms of vertical transmission and the reported preferential selection of escape mutants.

5. The study focuses on bnAb neutralization. How might other effector functions described for some of the bNabs (e.g., Fc-receptor mediated) contribute to immune clearance even when reduced neutralization is observed?

**Part II – Major Issues: Key Experiments Required for Acceptance**

Reviewer #1: Major concerns:

What is the evidence that the viruses from the FRESH and Baby Cure cohorts are confidently TF viruses? Is it solely based on time of virus extraction, ie within 48 hours after birth in babies and median of 1-day post plasma viremia in adults? It would be helpful for the authors to outline the criteria for defining TF viruses in this study; perhaps a section in the methods where the readers can access this information and make their own conclusions based on this information. If there are uncertainties in classification of these viruses, the authors should also outline them in a new section of the methods (this is somewhat addressed in lines 607-613 but could be more fleshed out in the methods).

It would be more helpful to the readers if the authors revised figures 3 and 4 to include virus names, neutralization tiers (1/2/unknown) and clade/subtype. The PIDs or numbers shown are not informative. Additionally, there is an emerging discussion in the HIV field of whether the current panels of HIV strains used to assess breadth of bnAbs from adults are appropriate to assess bnAbs from pediatric populations. There are two bnAbs previously isolated from pediatric populations, BF520 and AIIMS_330; both of which were tested on different virus panels for breadth. There is an opportunity here for the authors to test BF520 and AIIMS_330 against the adult and infant TF viruses in this study as it would be helpful to the field to see whether the sensitivity of adult or infant TF viruses show differences in sensitivity to infant bnAbs. These results may also provide insights into whether current bnAbs from adults are adequate for treating infants living with HIV.

To better understand the results in lines 392-395, is there any evidence that the structures of Envs from TF viruses in infants have modifications or differences (CD4-binding site accessibility, etc) from those in adults, thus influencing the enhanced sensitivity to majority of the bnAbs by the baby TF viruses compared to those from adults? Maybe something for the authors to also speculate on based on information that can be ascertained from the viral Env sequencing if structural analysis is not possible.

The authors did a great job with sequencing analyses to inform bnAb sensitivity or resistance, but some key residues or regions were not mentioned and it is not clear whether these analyses were done and the results not informative or whether they were excluded. For example, the residues in the loop D, V5 loop and bridging sheet were studied, but not the CD4-binding loop; all of which comprises the CD4-binding site. Any evidence for conserved residues in the CD4-binding loop such as D368 or mutations at this site that would influence sensitivity or resistance to CD4BS antibodies? For V2-apex, no information provided on N156 in V2-apex. For V3 glycan sensitivity, it is becoming more clearer than Envs with short V1 or V1-glycan deletions are more sensitive to V3-glycan bnAbs due to access to the V3-glycan bnAb epitope, so these may be other areas to study to better understand sensitivity to V3-glycan bnAbs. Finally, in MPER, only distal MPER epitopes assessed that is relevant for 10E8 targeting, but proximal MPER Abs have been recently generated via vaccination in adult humans and SIV-infected macaques, so any information on sequence variability in this region may be helpful to fully understand how MPER bnAbs can be incorporated into neonatal responses to HIV.

For such a high impact study, the discussion section could be revised to contribute to ongoing discussions in the field of pediatric HIV immunity. First, lines 553-605 in some part were already reported in the results section and could be significantly reduced to make the main points of sensitivity to the different bnAbs. Second, as the authors appropriately mentioned, studying the moms in the Baby Cure cohort could provide insights into the potential mechanisms of viral transmission/ dynamics of virus sensitivity to bnAbs from adults to infants. If virus is more resistant to bnAbs in adults but more sensitive to said bnAbs in infants, is it that moms harbor a mixture of viruses with different sensitivities to bnAbs and only the bnAb-sensitive ones get transmitted or is there some biological factors at play to alter virus bnAb-sensitivity in utero or soon after birth? The authors could speculate on some of these concepts in the discussion. Third, these results may provide implications for HIV vaccine designs in pediatric populations. For instance, do the authors see any value in these Envs with sensitivity to bnAbs serving as appropriate immunogens in future HIV vaccine strategies? These are all areas of interest to the field that could be incorporated into the discussion section.

Reviewer #2: n/a

Reviewer #3: One concern is how representative the env used for these studies are of T/F viruses. For example, a median of only 6 SGA env were generated for study of FRESH cohort samples ; it would be interesting to see how representative the SGA env used in the assays were of the viral population present at the time of sampling. Similarly, single infant env derived from bulk PCR were evaluated despite very high viral loads in many infants.

**Part III – Minor Issues: Editorial and Data Presentation Modifications**

Reviewer #1: Additional minor concerns:

1. Update language throughout manuscript from HIV-1-infected to people with/ living with HIV. Eg. lines 136, 138, 273, 276 etc.

2. To avoid confusion, whenever the authors reference “mothers”, they should consider indicating mothers for infants in the Baby Cure cohort, for eg. lines 286-287.

3. Treatment regimen at bottom of table 2; how does this link to the table? More information needed here, including symbols?

4. Figure 1 legend should include timepoints of samples for viral load measurements and length of time on ART. This information would provide clarity to the readers when reviewing these data.

Reviewer #2: Stats in fig 1 don’t include infants overall and mom overall? Were the medians tested and found non-significant?

Fig 1B and C is mistakenly described as Fig 2B and C in the results section. Not super clear in the figure legend what the difference is between Fig 1A and 1B or why transmitting mothers would be different from the FRESH cohort?

Reviewer #3: (No Response)

PLOS authors have the option to publish the peer review history of their article (what does this mean? ). If published, this will include your full peer review and any attached files.

**Do you want your identity to be public for this peer review?** For information about this choice, including consent withdrawal, please see our Privacy Policy .

Reviewer #1: No

Reviewer #2: No

Reviewer #3: No

**Figure resubmission:**

**Reproducibility:**



---

## [Decision Letter · Decision Letter 1]

Dear Prof Ndung’u,

We are pleased to inform you that your manuscript 'Distinct neutralization sensitivity between adult and infant transmitted/ founder HIV-1 subtype C viruses to broadly neutralizing monoclonal antibodies' has been provisionally accepted for publication in PLOS Pathogens.

Best regards,

R. Keith Reeves

Academic Editor

PLOS Pathogens

Susan Ross

Section Editor

PLOS Pathogens

Sumita Bhaduri-McIntosh

Editor-in-Chief

PLOS Pathogens

orcid.org/0000-0003-2946-9497

Michael Malim

Editor-in-Chief

PLOS Pathogens

orcid.org/0000-0002-7699-2064

Reviewer Comments (if any, and for reference):

Reviewer's Responses to Questions

**Part I - Summary**

Reviewer #1: Very elegant study with high translational potential. The results presented in this revised manuscript may inform future therapeutic strategies for infants and children with HIV. The authors did an excellent job in addressing the comments and critiques from the first round of interviews.

Reviewer #2: The authors were responsive to the reviews.

**Part II – Major Issues: Key Experiments Required for Acceptance**

Reviewer #1: Given the focus of the study on env sequence analyses, it may be beneficial to the scientific community to make these sequences available via a public database or provide a statement on data accessibility in this regard; ie, the 383 env sequences from 39 adult participants, and 41 env sequences from 21 infants. For confidentiality, the authors could consider de-identifying the patient information in the sequence IDs. I appreciate that the logo plots in figure 6 is a summary of the analyses for these sequences, thus at least a statement of env sequence accessibility if needed by a scientific researcher would suffice.

Reviewer #2: N/a

**Part III – Minor Issues: Editorial and Data Presentation Modifications**

Reviewer #1: 1. Figures were low resolution so poor quality to review; ensure that this is corrected with the journal for publication.

2. Since tier phenotyping for neutralization sensitivity or resistance is not available for the viruses tested, the authors should make this point in the legend for Figure 3 or in the methods section that described the neutralization assay.

3. Line 469 should say "Env" for protein structure, and not italicized env indicative of the gene.

4. It should be Table 1, not 1A unless I missed Table 1B - page 15 (between lines 350-351)

5. Line 534 needs attention - missing a semi-colon?

Reviewer #2: N/a

PLOS authors have the option to publish the peer review history of their article (what does this mean? ). If published, this will include your full peer review and any attached files.

**Do you want your identity to be public for this peer review?** For information about this choice, including consent withdrawal, please see our Privacy Policy .

Reviewer #1: No

Reviewer #2: No

---

## [Editor Report · Acceptance letter]

Dear Prof Ndung’u,

We are delighted to inform you that your manuscript, "Distinct neutralization sensitivity between adult and infant transmitted/ founder HIV-1 subtype C viruses to broadly neutralizing monoclonal antibodies," has been formally accepted for publication in PLOS Pathogens.

Best regards,

Sumita Bhaduri-McIntosh

Editor-in-Chief

PLOS Pathogens

orcid.org/0000-0003-2946-9497

Michael Malim

Editor-in-Chief

PLOS Pathogens

orcid.org/0000-0002-7699-2064